# Stability of Impaired Humoral Immunity HIV-1 Models with Active and Latent Cellular Infections

**Noura H. AlShamrani** [1,*], **Reham H. Halawani** [1], **Wafa Shammakh** [1] and **Ahmed M. Elaiw** [2]

1   Department of Mathematics, Faculty of Science, University of Jeddah, P.O. Box 80327,
    Jeddah 21589, Saudi Arabia; rhalawani0020.stu@uj.edu.sa (R.H.H.); wmshammakh@uj.edu.sa (W.S.)
2   Department of Mathematics, Faculty of Science, King Abdulaziz University, P.O. Box 80203,
    Jeddah 21589, Saudi Arabia; aelaiwksu.edu.sa@kau.edu.sa
*   Correspondence: nhalshamrani@uj.edu.sa

**Abstract:** This research aims to formulate and analyze two mathematical models describing the within-host dynamics of human immunodeficiency virus type-1 (HIV-1) in case of impaired humoral immunity. These models consist of five compartments, including healthy $CD4^+T$ cells, (HIV-1)-latently infected cells, (HIV-1)-actively infected cells, HIV-1 particles, and B-cells. We make the assumption that healthy cells can become infected when exposed to: (i) HIV-1 particles resulting from viral infection (VI), (ii) (HIV-1)-latently infected cells due to latent cellular infection (CI), and (iii) (HIV-1)-actively infected cells due to active CI. In the second model, we introduce distributed time-delays. For each of these systems, we demonstrate the non-negativity and boundedness of the solutions, calculate the basic reproductive number, identify all possible equilibrium states, and establish the global asymptotic stability of these equilibria. We employ the Lyapunov method in combination with LaSalle's invariance principle to investigate the global stability of these equilibrium points. Theoretical findings are subsequently validated through numerical simulations. Additionally, we explore the impact of B-cell impairment, time-delays, and CI on HIV-1 dynamics. Our results indicate that weakened immunity significantly contributes to disease progression. Furthermore, the presence of time-delays can markedly decrease the basic reproductive number, thereby suppressing HIV-1 replication. Conversely, the existence of latent CI spread increases the basic reproductive number, intensifying the progression of HIV-1. Consequently, neglecting latent CI spread in the HIV-1 dynamics model can lead to an underestimation of the basic reproductive number, potentially resulting in inaccurate or insufficient drug therapies for eradicating HIV-1 from the body. These findings offer valuable insights that can enhance the understanding of HIV-1 dynamics within a host.

**Keywords:** HIV-1; cellular infection; latently infected cells; immune impairment; global stability; distributed delays; Lyapunov function

**MSC:** 34D20; 34D23; 37N25; 92B05





## 1. Introduction

Human immunodeficiency virus type-1 (HIV-1), a retrovirus, targets $CD4^+T$ cells, a pivotal component of the immune system. In a healthy individual, the concentration of $CD4^+T$ cells is typically around 1000 cells/mm$^3$. However, following infection, a gradual decline in $CD4^+T$ cell count sets in, and this decrease may persist for years. Whenever the concentration of these cells falls below 200 cells/mm$^3$, the individual is diagnosed with Acquired Immune Deficiency Syndrome (AIDS) as indicated by [1]. According to the World Health Organization's report at the close of 2022, approximately 39 million individuals were living with HIV-1 worldwide [2]. The adaptive immune response has an effective role in resisting and fighting viruses that attack the human body. B-cells and cytotoxic T lymphocytes (CTLs) represent two crucial components of the adaptive immune response. B-cells produce antibodies to counteract HIV-1 particles, while CTLs

eliminate cells that have been infected by HIV-1. The assessment of interactions between HIV-1 and both target cells and immune cells can incur significant experimental costs. Consequently, mathematical modeling of HIV-1 infection has become an indispensable tool for comprehending the dynamic behavior of HIV-1 particles and their interactions with target cells and immune cells. The virus dynamics model under the effect of humoral immunity, was introduced in [3] as follows:

Healthy CD4$^+$T cells: $\quad \dfrac{dF(t)}{dt} = \underbrace{\psi}_{\text{Production of healthy cells}} - \underbrace{\rho F(t)}_{\text{Natural death}} - \underbrace{\varphi F(t)K(t)}_{\text{Infectious transmission}},$ (1)

(HIV-1)-actively infected cells: $\quad \dfrac{dG(t)}{dt} = \underbrace{\varphi F(t)K(t)}_{\text{Infectious transmission}} - \underbrace{\theta G(t)}_{\text{Natural death}},$ (2)

Free HIV-1 particles: $\quad \dfrac{dK(t)}{dt} = \underbrace{\mu G(t)}_{\text{Burst size}} - \underbrace{\kappa K(t)}_{\text{Natural death}} - \underbrace{\sigma U(t)K(t)}_{\text{Neutralization of HIV-1}},$ (3)

B-cells: $\quad \dfrac{dU(t)}{dt} = \underbrace{\nu K(t)U(t)}_{\text{B-cell stimulation}} - \underbrace{\gamma U(t)}_{\text{Natural death}},$ (4)

where, $F(t)$, $G(t)$, $K(t)$, and $U(t)$ represent the concentrations of healthy CD4$^+$T cells, (HIV-1)-actively infected cells, free HIV-1 particles, and B-cells at time $t$. The parameters $\psi$, $\varphi$, $\mu$, $\sigma$, and $\nu$ correspond to the production rate of healthy CD4$^+$T cells, incidence (infection) rate, generation rate of HIV-1 from infected cells, neutralization rate of HIV-1 by B-cells, and stimulation rate of the B-cells, respectively. The death rate constants for the four compartments $F$, $G$, $K$, and $U$ are denoted by $\rho$, $\theta$, $\kappa$, and $\gamma$, respectively. In this model, it is assumed that HIV-1 particles attack and infect healthy CD4$^+$T cells. Once the healthy CD4$^+$T cells become infected, they begin to produce numerous HIV-1 particles. Simultaneously, the B-cells become activated and produce specific antibodies to neutralize the HIV-1 particles. Several extensions were made on this model by involving (i) (HIV-1)-latently infected cells [4], (ii) time-delay [5–7], (iii) diffusion [8,9], and (iv) age structure [10–12]. The B-cell dynamics can be written as:

$$\frac{dU(t)}{dt} = \Theta(K(t), U(t)) - \gamma U(t),$$

where $\Theta(K, U)$ is stimulation of the B-cells. The literature has considered various special forms of $\Theta(K, U)$, including:

**Form-1.** One of the special forms of $\Theta(K, U)$ is the self-regulating humoral response, represented as $\Theta(K, U) = \omega$, where $\omega > 0$ as discussed in [13].

**Form-2.** Linear humoral response, $\Theta(K, U) = \phi K$, where $\phi > 0$ as cited in [14–16].

**Form-3.** Another form of $\Theta(K, U)$ is akin to a predator-prey interaction in the humoral response, expressed as $\Theta(K, U) = \nu KU$, where $\nu > 0$ as documented in several references including [5–7,13–15].

**Form-4.** A combination of Form-1, Form-2, and Form-3 for $\Theta(K, U)$ is represented as $\Theta(K, U) = \omega + \phi K + \nu KU$, as described in [13].

**Form-5.** The saturated humoral expansion is characterized by $\Theta(K, U) = \frac{\nu KU}{\varpi + U}$, where $\varpi > 0$, as discussed in [10,12].

Model (1)–(4) assumes that HIV-1 infection occurs exclusively through viral infection (VI) contact. However, extensive research findings have indicated an alternative route wherein HIV-1 can be directly transmitted from an infected CD4$^+$T cell to a healthy CD4$^+$T cell, facilitated by the formation of virological synapses [17–22]. This mode of infection, known as cellular infection (CI), exerts a profound impact on HIV-1 transmission, potentially being 100–1000 times more rapid than VI dissemination [23]. Previous investigations have delved into viral infection systems that encompass CI and humoral immunity, with no-

table studies found in [16,24,25]. In the papers [16,24,25], it was postulated that the presence of antigens solely stimulates the humoral immune response while disregarding any impairment in humoral immunity. According to a report in [26], HIV-1 has the potential to induce impairment in B-cells. When accounting for impaired humoral immunity, the population dynamics of B-cells can be described as shown in various references, including [27,28].

$$\frac{dU(t)}{dt} = \phi K(t) - \gamma U(t) - \varepsilon U(t)K(t),$$

where, $\phi K$ is stimulation of B-cells and $\varepsilon UK$ is the B-cells impairment and $\phi$ and $\varepsilon$ are constants.

In [27,28], the CI was not considered. Recently, Elaiw and Alshehaiween [29] and Elaiw et al. [30] introduced and analyzed virus dynamics models with impaired humoral immunity and CI. The models presented in [29,30] did not take into account the presence of (HIV-1)-latently infected cells. However, Miao et al. in [31] introduced viral infection models that incorporated humoral immunity, (HIV-1)-latently infected cells, and CI. Additionally, in the studies [32,33], the global stability of viral models was investigated, considering factors such as CI, multi-stages of infected cells, and adaptive immunity. In the works [31–33], it was assumed that CI solely resulted from the activities of (HIV-1)-actively infected cells. However, it was reported in [34] that (HIV-1)-latently infected cells can also participate in infecting healthy CD4$^+$T cells through the CI mechanism. Furthermore, in papers such as [35–38], various virus dynamics models were developed with the consideration that both (HIV-1)-latently and (HIV-1)-actively infected cells contribute to the CI mechanism. However, it's important to note that these papers did not account for humoral immune impairment.

The objective of this study is to introduce two within-host HIV-1 models that incorporate (HIV-1)-latently infected cells, humoral immune impairment, and the mechanism of CI. In these models, both (HIV-1)-latently infected and (HIV-1)-actively infected cells are considered contributors to CI. The second model further extends the analysis by incorporating three types of distributed time-delays. We perform a comprehensive analysis of both models, including establishing non-negativity and boundedness of solutions, calculating the basic reproductive number, identifying equilibria, assessing global stability, conducting numerical simulations, and engaging in a detailed discussion of the obtained results.

## 2. Model Incorporating Impaired Humoral Immunity and CI

### 2.1. Description of the System

We introduce an HIV-1 model that incorporates impaired humoral immunity, considering the infection of healthy CD4$^+$T cells upon contact with HIV-1 particles, (HIV-1)-latently infected cells, or (HIV-1)-actively infected cells. The model is presented as follows:

$$\begin{cases} \frac{dF(t)}{dt} = \psi - \rho F(t) - \varphi_1 F(t)K(t) - \varphi_2 F(t)Q(t) - \varphi_3 F(t)G(t), \\ \frac{dQ(t)}{dt} = \varphi_1 F(t)K(t) + \varphi_2 F(t)Q(t) + \varphi_3 F(t)G(t) - (\xi + \vartheta)Q(t), \\ \frac{dG(t)}{dt} = \xi Q(t) - \theta G(t), \\ \frac{dK(t)}{dt} = \mu G(t) - \kappa K(t) - \sigma U(t)K(t), \\ \frac{dU(t)}{dt} = \phi K(t) - \gamma U(t) - \varepsilon U(t)K(t), \end{cases} \tag{5}$$

where $Q(t)$, denotes the concentration of (HIV-1)-latently infected cells at time $t$. (HIV-1)-latently infected cells are activated at rate $\xi Q$ and die at rate $\vartheta Q$. In this context, $\varphi_1 FK$, $\varphi_2 FQ$, and $\varphi_3 FG$ represent the infection rates via HIV-1 particles (VI), (HIV-1)-latently infected cells (CI), and (HIV-1)-actively infected cells (CI), respectively. It is important to note that all these parameters are positive.

*2.2. Main Basic Properties*

2.2.1. Maintaining Non-Negativity and Boundedness in the Solutions

**Lemma 1.** *Consider the system (5), there exist positive constants $\tau_i > 0$, $i = 1, 2, 3$, such that the set*

$$\Omega = \left\{ (F, Q, G, K, U) \in \mathbb{R}^5_{\geq 0} : 0 \leq F(t), Q(t), G(t) \leq \tau_1, 0 \leq K(t) \leq \tau_2, 0 \leq U(t) \leq \tau_3 \right\}$$

*is positively invariant.*

**Proof.** To address the nonnegativity of solutions, from system (5) we have

$$\frac{dF}{dt} \mid_{F=0} = \psi > 0,$$

$$\frac{dQ}{dt} \mid_{Q=0} = \varphi_1 FK + \varphi_3 FG \geq 0, \text{ for all } F, K, G \geq 0,$$

$$\frac{dG}{dt} \mid_{G=0} = \xi Q \geq 0, \text{ for all } Q \geq 0,$$

$$\frac{dK}{dt} \mid_{K=0} = \mu G \geq 0, \text{ for all } G \geq 0,$$

$$\frac{dU}{dt} \mid_{U=0} = \phi K \geq 0, \text{ for all } K \geq 0.$$

Hence, $(F(t), Q(t), G(t), K(t), U(t)) \in \mathbb{R}^5_{\geq 0}$, for all $t \geq 0$ when $(F(0), Q(0), G(0), K(0), U(0)) \in \mathbb{R}^5_{\geq 0}$. Let

$$Y(t) = F(t) + Q(t) + G(t) + \frac{\theta}{2\mu} K(t) + \frac{\theta \kappa}{4\mu\phi} U(t).$$

Then, we have

$$\frac{dY(t)}{dt} = \frac{dF(t)}{dt} + \frac{dQ(t)}{dt} + \frac{dG(t)}{dt} + \frac{\theta}{2\mu} \frac{dK(t)}{dt} + \frac{\theta\kappa}{4\mu\phi} \frac{dU(t)}{dt}$$

$$= \psi - \rho F(t) - \vartheta Q(t) - \frac{\theta}{2} G(t) - \frac{\theta\kappa}{4\mu} K(t) - \left( \frac{\theta\sigma}{2\mu} + \frac{\theta\kappa\varepsilon}{4\mu\phi} \right) U(t)K(t) - \frac{\theta\kappa\gamma}{4\mu\phi} U(t)$$

$$\leq \psi - \rho F(t) - \vartheta Q(t) - \frac{\theta}{2} G(t) - \frac{\theta\kappa}{4\mu} K(t) - \frac{\theta\kappa\gamma}{4\mu\phi} U(t)$$

$$\leq \psi - \eta \left( F(t) + Q(t) + G(t) + \frac{\theta}{2\mu} K(t) + \frac{\theta\kappa}{4\mu\phi} U(t) \right) = \psi - \eta Y(t),$$

where $\eta = \min\left\{ \rho, \vartheta, \frac{\theta}{2}, \frac{\kappa}{2}, \gamma \right\}$. Hence,

$$Y(t) \leq e^{-\eta t} \left( Y(0) - \frac{\psi}{\eta} \right) + \frac{\psi}{\eta}.$$

This yields $0 \leq Y(t) \leq \tau_1$ if $Y(0) \leq \tau_1$, where $\tau_1 = \frac{\psi}{\eta}$.

Given that all state variables are non-negative, $0 \leq F(t), Q(t), G(t) \leq \tau_1, 0 \leq K(t) \leq \tau_2$, and $0 \leq U(t) \leq \tau_3$, for all $t \geq 0$ if $F(0) + Q(0) + G(0) + \frac{\theta}{2\mu} K(0) + \frac{\theta\kappa}{4\mu\phi} U(0) \leq \tau_1$, where $\tau_2 = \frac{2\mu\tau_1}{\theta}$ and $\tau_3 = \frac{4\mu\phi\tau_1}{\theta\kappa}$. Hence, $F(t)$, $Q(t)$, $G(t)$, $K(t)$, and $U(t)$ are all bounded, indicating that $\Omega$ is a positively invariant and compact set with respect to the system (5).  □

2.2.2. Analysis of Reproductive Numbers and Equilibrium Points

**Lemma 2.** *There exists a basic reproductive number* $\Re_0 = \frac{F_0(\varphi_1\mu\xi+\varphi_2\kappa\theta+\varphi_3\xi\kappa)}{(\xi+\vartheta)\kappa\theta}$ *for system* (5), *such that*

(i)     *the system always has an infection-free equilibrium point* $\Xi Q_0$, *and*
(ii)    *if* $\Re_0 > 1$, *the system also has an infected equilibrium point* $\Xi Q_1$.

**Proof.** It is evident that system (5) invariably possesses an infection-free equilibrium denoted as $\Xi Q_0 = (F_0, 0, 0, 0, 0)$, where $F_0 = \frac{\psi}{\rho}$. In the subsequent analysis, we will employ the next-generation matrix method proposed by Driessche and Watmough [39] to compute the basic reproductive number for system (5). Considering the infected compartments in model (5), arranged as $(Q, G, K)$, the nonlinear terms involving the new infection $\hat{\Gamma}_1$ and the outflow term $\hat{\Delta}_1$ are represented by the following matrices:

$$\hat{\Gamma}_1 = \begin{pmatrix} \varphi_1 FK + \varphi_2 FQ + \varphi_3 FG \\ 0 \\ 0 \end{pmatrix}, \quad \hat{\Delta}_1 = \begin{pmatrix} (\xi+\vartheta)Q \\ -\xi Q + \theta\, G \\ -\mu G + \kappa K + \sigma UK \end{pmatrix}.$$

We calculate the derivatives of $\hat{\Gamma}_1$ and $\hat{\Delta}_1$ at the equilibrium $\Xi Q_0$, resulting in the following matrices:

$$\Gamma_1 = \begin{pmatrix} \varphi_2 F_0 & \varphi_3 F_0 & \varphi_1 F_0 \\ 0 & 0 & 0 \\ 0 & 0 & 0 \end{pmatrix}, \quad \Delta_1 = \begin{pmatrix} \xi+\vartheta & 0 & 0 \\ -\xi & \theta & 0 \\ 0 & -\mu & \kappa \end{pmatrix}.$$

The next-generation matrix takes the following form:

$$\Gamma_1\Delta_1^{-1} = \begin{pmatrix} \frac{F_0(\varphi_1\mu\xi+\varphi_2\kappa\theta+\varphi_3\xi\kappa)}{(\xi+\vartheta)\kappa\theta} & \frac{F_0(\varphi_1\mu+\varphi_3\kappa)}{\kappa\theta} & \frac{\varphi_1 F_0}{\kappa} \\ 0 & 0 & 0 \\ 0 & 0 & 0 \end{pmatrix}.$$

The basic reproductive number $\Re_0$ is defined as the spectral radius of the matrix $\Gamma_1\Delta_1^{-1}$ and is calculated as follows:

$$\Re_0 = \frac{F_0(\varphi_1\mu\xi + \varphi_2\kappa\theta + \varphi_3\xi\kappa)}{(\xi+\vartheta)\kappa\theta} = \Re_{01} + \Re_{02} + \Re_{03}, \tag{6}$$

where

$$\Re_{01} = \frac{F_0\mu\xi\varphi_1}{\kappa\theta(\xi+\vartheta)}, \qquad \Re_{02} = \frac{F_0\varphi_2}{\xi+\vartheta}, \qquad \Re_{03} = \frac{F_0\xi\varphi_3}{\theta(\xi+\vartheta)}.$$

The parameter $\Re_0$ holds significant clinical relevance, as it determines whether the HIV-1 infection will become chronic or not. In this context, $\Re_{01}$, $\Re_{02}$, and $\Re_{03}$ represent the average numbers of secondary infected cells resulting from contacts with HIV-1 particles, latently infected cells, and actively infected cells, respectively. To identify additional equilibria apart from $\Xi Q_0$, we consider $(F, Q, G, K, U)$ as any equilibrium satisfying the following algebraic equations:

$$\psi - \rho F - \varphi_1 FK - \varphi_2 FQ - \varphi_3 FG = 0, \tag{7}$$
$$\varphi_1 FK + \varphi_2 FQ + \varphi_3 FG - (\xi+\vartheta)Q = 0, \tag{8}$$
$$\xi Q - \theta G = 0, \tag{9}$$
$$\mu G - \kappa K - \sigma UK = 0, \tag{10}$$
$$\phi K - \gamma U - \varepsilon UK = 0. \tag{11}$$

From Equations (9) and (11), we get

$$Q = \frac{\theta G}{\xi}, \quad U = \frac{\phi K}{\gamma + \varepsilon K}. \tag{12}$$

Upon substitution of Equation (12) into Equation (10), the result is as follows:

$$G = \frac{\gamma \kappa K + (\sigma \phi + \kappa \varepsilon)K^2}{\mu(\gamma + \varepsilon K)}. \tag{13}$$

Upon substitution of Equation (13) into Equation (12), the result is as follows:

$$Q = \frac{\theta\left(\gamma \kappa K + (\sigma \phi + \kappa \varepsilon)K^2\right)}{\xi \mu(\gamma + \varepsilon K)}. \tag{14}$$

From Equations (7) and (8), we get

$$\psi - \rho F = (\xi + \vartheta)Q. \tag{15}$$

Upon substitution of Equation (14) into Equation (15), the result is as follows:

$$F = \frac{1}{\rho}\left(\psi - \frac{\theta(\xi + \vartheta)\left(\gamma \kappa K + (\sigma \phi + \kappa \varepsilon)K^2\right)}{\xi \mu(\gamma + \varepsilon K)}\right). \tag{16}$$

Upon substitution of Equations (13), (14) and (16) into Equation (8), the result is as follows:

$$\frac{K}{\rho \mu^2 \xi^2 (\gamma + \varepsilon K)^2}\left(AK^3 + BK^2 + CK + D\right) = 0, \tag{17}$$

where

$$A = \theta(\xi + \vartheta)(\sigma \phi + \varepsilon \kappa)(\varepsilon \xi \mu \varphi_1 + (\sigma \phi + \varepsilon \kappa)(\theta \varphi_2 + \xi \varphi_3)),$$
$$B = \mu \xi \varphi_1\left(\gamma \theta(\xi + \vartheta)(\sigma \phi + 2\varepsilon \kappa) - \psi \mu \xi \varepsilon^2\right)$$
$$\quad + (\sigma \phi + \varepsilon \kappa)(\varepsilon \mu \rho \xi \theta(\xi + \vartheta) - (\psi \varepsilon \mu \xi - 2\gamma \kappa \theta(\xi + \vartheta))(\theta \varphi_2 + \xi \varphi_3)),$$
$$C = \gamma(\rho \mu \xi \theta(\xi + \vartheta)(\sigma \phi + 2\varepsilon \kappa) + \mu \xi \varphi_1(\gamma \kappa \theta(\xi + \vartheta) - 2\psi \varepsilon \mu \xi)$$
$$\quad - \left(\psi \mu \xi(\sigma \phi + 2\varepsilon \kappa) - \gamma \theta \kappa^2(\xi + \vartheta)\right)(\theta \varphi_2 + \xi \varphi_3)),$$
$$D = \mu \rho \kappa \xi \theta \gamma^2(\xi + \vartheta)(1 - \Re_0),$$

where $\Re_0$ is given by Equation (6). From Equation (17), we have

1.  If $K = 0$, then from Equations (12)–(14) and (16) the infection-free equilibrium $\Xi Q_0$ is obtained.
2.  If $K \neq 0$, we have the equation $AK^3 + BK^2 + CK + D = 0$. In this scenario, let us introduce a function $\Psi(K)$ defined on the interval $[0, \infty)$ as:

$$\Psi(K) = AK^3 + BK^2 + CK + D.$$

Then

$$\Psi(0) = \mu \rho \kappa \xi \theta \gamma^2(\xi + \vartheta)(1 - \Re_0) < 0, \text{ if } \Re_0 > 1,$$
$$\lim_{K \to \infty} \Psi(K) = \infty.$$

As a result, the function $\Psi$ possesses a positive real root denoted as $K_1$. Consequently, by substituting Equations (13) and (14) into Equation (7), we obtain:

$$F_1 = \frac{\psi}{\rho + \varphi_1 K_1 + \varphi_2 Q_1 + \varphi_3 G_1},$$

where

$$Q_1 = \frac{\theta \left( \gamma \kappa K_1 + (\sigma \phi + \kappa \varepsilon) K_1^2 \right)}{\xi \mu (\gamma + \varepsilon K_1)}, \quad G_1 = \frac{\gamma \kappa K_1 + (\sigma \phi + \kappa \varepsilon) K_1^2}{\mu (\gamma + \varepsilon K_1)}, \quad U_1 = \frac{\phi K_1}{\gamma + \varepsilon K_1}.$$

The existence of the infected equilibrium $\Xi Q_1 = (F_1, Q_1, G_1, K_1, U_1)$ is evident when $\Re_0 > 1$.  □

### 2.2.3. The Analysis of the Stability of the Equilibria $\Xi\,Q_0$ and $\Xi\,Q_1$

**Theorem 1.** *If $\Re_0 < 1$, then the equilibrium $\Xi Q_0$ of system (5) is locally asymptotically stable (L.A.S), and it becomes unstable when $\Re_0 > 1$.*

**Proof.** In accordance with the study proposed by Willems [40], the local asymptotic stability of the equilibrium $\Xi Q_0$ can be ascertained by examining the eigenvalues of its associated Jacobian matrix, which is presented as follows:

$$J = \begin{pmatrix} -\rho - \varphi_1 K - \varphi_2 Q - \varphi_3 G & -\varphi_2 F & -\varphi_3 F & -\varphi_1 F & 0 \\ \varphi_1 K + \varphi_2 Q + \varphi_3 G & \varphi_2 F - (\vartheta + \xi) & \varphi_3 F & \varphi_1 F & 0 \\ 0 & \xi & -\theta & 0 & 0 \\ 0 & 0 & \mu & -(\kappa + \sigma U) & -\sigma K \\ 0 & 0 & 0 & \phi - \varepsilon U & -(\gamma + \varepsilon K) \end{pmatrix}. \tag{18}$$

At the *infection-free equilibrium point* $\Xi Q_0$ the Jacobian matrix becomes

$$J_{\Xi Q_0} = \begin{pmatrix} -\rho & -\varphi_2 F_0 & -\varphi_3 F_0 & -\varphi_1 F_0 & 0 \\ 0 & \varphi_2 F_0 - (\vartheta + \xi) & \varphi_3 F_0 & \varphi_1 F_0 & 0 \\ 0 & \xi & -\theta & 0 & 0 \\ 0 & 0 & \mu & -\kappa & 0 \\ 0 & 0 & 0 & \phi & -\gamma \end{pmatrix}. \tag{19}$$

For matrix (19), the characteristic equation $\left| J_{\Xi Q_0} - x I_5 \right| = 0$ is solved as $(x + \gamma)(x + \rho)Y(x) = 0$, where $x$ is the eigenvalue, $I_5$ is the identity matrix and

$$Y(x) = x^3 + m_2 x^2 + m_1 x + m_0, \tag{20}$$

and

$$m_0 = \kappa \theta (\xi + \vartheta)(1 - \Re_0) > 0,$$
$$m_1 = \kappa \theta + \kappa (\xi + \vartheta)(1 - \Re_{02}) + \theta (\xi + \vartheta)(1 - (\Re_{02} + \Re_{03})) > 0,$$
$$m_2 = \kappa + \theta + (\xi + \vartheta)(1 - \Re_{02}) > 0,$$
$$m_1 m_2 - m_0 = \frac{\psi \mu \xi \varphi_1}{\rho} + (\theta + (\xi + \vartheta)(1 - \Re_{02}))(\kappa(\kappa + \theta) + \kappa(\xi + \vartheta)(1 - \Re_{02})$$
$$+ \theta (\xi + \vartheta)(1 - (\Re_{02} + \Re_{03}))) > 0,$$

where $\Re_0 < 1$.

The Jacobian matrix $J_{\Xi Q_0}$ is evidently characterized by two negative eigenvalues, $-\gamma$ and $-\rho$. The remaining eigenvalues are determined as the solutions to the cubic equation presented in (20). Applying the Routh-Hurwitz criteria [40], it is apparent that all roots of Equation (20) possess negative real parts. Consequently, the infection-free equilibrium $\Xi Q_0$ is locally asymptotically stable (L.A.S) when $\Re_0 < 1$. Conversely, if $\Re_0 > 1$, we have

$m_0 < 0$, implying that Equation (20) must have at least one positive real root. Hence, the equilibrium $\Xi Q_0$ becomes unstable when $\Re_0 > 1$. $\square$

In the upcoming theorems, we will delve into the global stability of the equilibria. The construction of Lyapunov function is formulated following the method demonstrated in [41]. We defined $H(z) = z - 1 - \ln(z)$ and denote $(F, Q, G, K, U) = (F(t), Q(t), G(t), K(t), U(t))$.

**Theorem 2.** *For system* (5), *if* $\Re_0 < 1$, *then* $\Xi Q_0$ *is globally asymptotically stable (G.A.S).*

**Proof.** Let's consider a candidate Lyapunov function

$$\Theta_0 = F_0 H\left(\frac{F}{F_0}\right) + Q + \frac{(\xi + \vartheta)(1 - \Re_{02})}{\xi} G + \frac{\theta(\xi + \vartheta)(1 - (\Re_{02} + \Re_{03}))}{\xi\mu} K + \frac{\theta\kappa(\xi + \vartheta)(1 - \Re_0)}{\phi\xi\mu} U.$$

Clearly, $\Theta_0(F, Q, G, K, U) > 0$ for all $F, Q, G, K, U > 0$, and $\Theta_0(F_0, 0, 0, 0, 0) = 0$. Computing $\frac{d\Theta_0}{dt}$ along with the solutions of model (5), we get

$$\frac{d\Theta_0}{dt} = \left(1 - \frac{F_0}{F}\right)\frac{dF}{dt} + \frac{dQ}{dt} + \frac{(\xi + \vartheta)(1 - \Re_{02})}{\xi}\frac{dG}{dt} + \frac{\theta(\xi + \vartheta)(1 - (\Re_{02} + \Re_{03}))}{\xi\mu}\frac{dK}{dt}$$

$$+ \frac{\theta\kappa(\xi + \vartheta)(1 - \Re_0)}{\phi\xi\mu}\frac{dU}{dt}$$

$$= \left(1 - \frac{F_0}{F}\right)(\psi - \rho F - \varphi_1 FK - \varphi_2 FQ - \varphi_3 FG) + \varphi_1 FK + \varphi_2 FQ + \varphi_3 FG - (\xi + \vartheta)Q$$

$$+ \frac{(\xi + \vartheta)(1 - \Re_{02})}{\xi}(\xi Q - \theta G) + \frac{\theta(\xi + \vartheta)(1 - (\Re_{02} + \Re_{03}))}{\xi\mu}(\mu G - \kappa K - \sigma UK)$$

$$+ \frac{\theta\kappa(\xi + \vartheta)(1 - \Re_0)}{\phi\xi\mu}(\phi K - \gamma U - \varepsilon UK)$$

$$= \left(1 - \frac{F_0}{F}\right)(\psi - \rho F) + (\varphi_2 F_0 - (\xi + \vartheta) + (\xi + \vartheta)(1 - \Re_{02}))Q$$

$$+ \left(\varphi_3 F_0 - \frac{\theta(\xi + \vartheta)(1 - \Re_{02})}{\xi} + \frac{\theta(\xi + \vartheta)(1 - (\Re_{02} + \Re_{03}))}{\xi}\right)G$$

$$+ \left(\varphi_1 F_0 - \frac{\theta\kappa(\xi + \vartheta)(1 - (\Re_{02} + \Re_{03}))}{\xi\mu} + \frac{\theta\kappa(\xi + \vartheta)(1 - \Re_0)}{\xi\mu}\right)K$$

$$- \frac{\gamma\theta\kappa(\xi + \vartheta)(1 - \Re_0)}{\phi\xi\mu}U - \frac{\theta(\xi + \vartheta)}{\phi\xi\mu}(\phi\sigma(1 - (\Re_{02} + \Re_{03})) + \varepsilon\kappa(1 - \Re_0))UK.$$

After performing a direct calculation and utilizing the value $F_0 = \psi/\rho$, we acquire:

$$\frac{d\Theta_0}{dt} = \left(1 - \frac{F_0}{F}\right)(\rho F_0 - \rho F) - \frac{\gamma\theta\kappa(\xi + \vartheta)(1 - \Re_0)}{\phi\xi\mu}U$$

$$- \frac{\theta(\xi + \vartheta)}{\phi\xi\mu}(\phi\sigma(1 - (\Re_{02} + \Re_{03})) + \varepsilon\kappa(1 - \Re_0))UK$$

$$= -\frac{\rho(F - F_0)^2}{F} - \frac{\gamma\theta\kappa(\xi + \vartheta)(1 - \Re_0)}{\phi\xi\mu}U$$

$$- \frac{\theta(\xi + \vartheta)}{\phi\xi\mu}(\phi\sigma(1 - (\Re_{02} + \Re_{03})) + \varepsilon\kappa(1 - \Re_0))UK.$$

Clearly, $\frac{d\Theta_0}{dt} \leq 0$ when $\Re_0 < 1$ and $\frac{d\Theta_0}{dt} = 0$ when $F = F_0$ and $K = U = 0$. Let $\Phi_0 = \left\{(F, Q, G, K, U) : \frac{d\Theta_0}{dt} = 0\right\}$, and considering the largest invariant subset of $\Phi_0$ as $\Phi_0'$. Hence,

all solutions converge to: $\Phi_0'$ [42]. In $\Phi_0'$, all elements satisfy $F(t) = F_0$ and $K(t) = U(t) = 0$. Subsequently, the fourth equation in system (5) yields

$$0 = \frac{dK(t)}{dt} = \mu G(t) \Longrightarrow G(t) = 0, \text{ for all } t.$$

Furthermore, the first equation in system (5) results in

$$0 = \frac{dF(t)}{dt} = \psi - \rho F_0 - \varphi_2 F_0 Q(t) \Longrightarrow Q(t) = 0, \text{ for all } t.$$

Therefore, $\Phi_0' = \{(F, Q, G, K, U) \in \Phi_0 : F = F_0, Q = G = K = U = 0\} = \{\Xi Q_0\}$. Hence, we conclude that $\Xi Q_0$ is G.A.S whenever $\Re_0 < 1$ based on LaSalle's invariance principle (L.I.P) [42]. $\square$

**Theorem 3.** *If $\Re_0 > 1$, the equilibrium $\Xi Q_1$ of the system (5) is G.A.S.*

**Proof.** We consider the function $\Theta_1(F, Q, G, K, U)$ given by Equation (21) as:

$$\Theta_1 = F_1 H\left(\frac{F}{F_1}\right) + Q_1 H\left(\frac{Q}{Q_1}\right) + \frac{F_1(\varphi_1\mu + \varphi_3(\kappa + \sigma U_1))G_1}{\theta(\kappa + \sigma U_1)} H\left(\frac{G}{G_1}\right) + \frac{\varphi_1 F_1 K_1}{\kappa + \sigma U_1} H\left(\frac{K}{K_1}\right) \tag{21}$$
$$+ \frac{\sigma \varphi_1 F_1}{2(\kappa + \sigma U_1)(\phi - \varepsilon U_1)}(U - U_1)^2.$$

Equilibrium condition Equation (11) guarantees that $\phi - \varepsilon U_1 = \dfrac{\gamma U_1}{K_1} > 0$. Clearly, $\Theta_1$ is positive definite. Calculating $\frac{d\Theta_1}{dt}$:

$$\frac{d\Theta_1}{dt} = \left(1 - \frac{F_1}{F}\right)\frac{dF}{dt} + \left(1 - \frac{Q_1}{Q}\right)\frac{dQ}{dt} + \frac{F_1(\varphi_1\mu + \varphi_3(\kappa + \sigma U_1))}{\theta(\kappa + \sigma U_1)}\left(1 - \frac{G_1}{G}\right)\frac{dG}{dt}$$
$$+ \frac{\varphi_1 F_1}{\kappa + \sigma U_1}\left(1 - \frac{K_1}{K}\right)\frac{dK}{dt} + \frac{\sigma \varphi_1 F_1}{(\kappa + \sigma U_1)(\phi - \varepsilon U_1)}(U - U_1)\frac{dU}{dt}$$
$$= \left(1 - \frac{F_1}{F}\right)(\psi - \rho F - \varphi_1 FK - \varphi_2 FQ - \varphi_3 FG) + \left(1 - \frac{Q_1}{Q}\right)(\varphi_1 FK + \varphi_2 FQ + \varphi_3 FG - (\xi + \vartheta)Q)$$
$$+ \frac{F_1(\varphi_1\mu + \varphi_3(\kappa + \sigma U_1))}{\theta(\kappa + \sigma U_1)}\left(1 - \frac{G_1}{G}\right)(\xi Q - \theta G) + \frac{\varphi_1 F_1}{\kappa + \sigma U_1}\left(1 - \frac{K_1}{K}\right)(\mu G - \kappa K - \sigma UK)$$
$$+ \frac{\sigma \varphi_1 F_1}{(\kappa + \sigma U_1)(\phi - \varepsilon U_1)}(U - U_1)(\phi K - \gamma U - \varepsilon UK)$$
$$= \left(1 - \frac{F_1}{F}\right)(\psi - \rho F) + \left(\varphi_2 F_1 - (\xi + \vartheta) + \frac{\xi F_1(\varphi_1\mu + \varphi_3(\kappa + \sigma U_1))}{\theta(\kappa + \sigma U_1)}\right)Q + \varphi_1 F_1 K$$
$$+ \left(\varphi_3 F_1 - \frac{F_1(\varphi_1\mu + \varphi_3(\kappa + \sigma U_1))}{\kappa + \sigma U_1} + \frac{\mu \varphi_1 F_1}{\kappa + \sigma U_1}\right)G - (\varphi_1 FK + \varphi_2 FQ + \varphi_3 FG)\frac{Q_1}{Q}$$
$$+ (\xi + \vartheta)Q_1 - \frac{\xi F_1(\varphi_1\mu + \varphi_3(\kappa + \sigma U_1))}{\theta(\kappa + \sigma U_1)}\frac{QG_1}{G} + \frac{F_1(\varphi_1\mu + \varphi_3(\kappa + \sigma U_1))G_1}{\kappa + \sigma U_1}$$
$$- \frac{\mu \varphi_1 F_1}{\kappa + \sigma U_1}\frac{GK_1}{K} - \frac{\kappa \varphi_1 F_1}{\kappa + \sigma U_1}(K - K_1) - \frac{\sigma \varphi_1 F_1}{\kappa + \sigma U_1}U(K - K_1)$$
$$+ \frac{\sigma \varphi_1 F_1}{(\kappa + \sigma U_1)(\phi - \varepsilon U_1)}(U - U_1)(\phi K - \gamma U - \varepsilon UK). \tag{22}$$

Using the following equilibrium conditions for $\Xi Q_1$

$$\psi = \rho F_1 + \varphi_1 F_1 K_1 + \varphi_2 F_1 Q_1 + \varphi_3 F_1 G_1,$$

$$\varphi_1 F_1 K_1 + \varphi_2 F_1 Q_1 + \varphi_3 F_1 G_1 = (\xi + \vartheta) Q_1,$$

$$Q_1 = \frac{\theta G_1}{\xi} \implies G_1 = \frac{\xi Q_1}{\theta},$$

$$G_1 = \frac{(\kappa + \sigma U_1) K_1}{\mu}, \qquad Q_1 = \frac{\theta(\kappa + \sigma U_1) K_1}{\xi \mu},$$

$$\phi K_1 = \gamma U_1 + \varepsilon U_1 K_1,$$

we get

$$\varphi_1 F_1 K_1 + \varphi_3 F_1 G_1 = \frac{F_1(\varphi_1 \mu + \varphi_3(\kappa + \sigma U_1)) K_1}{\mu} = \frac{\xi F_1(\varphi_1 \mu + \varphi_3(\kappa + \sigma U_1)) Q_1}{\theta(\kappa + \sigma U_1)},$$

$$\left( \varphi_2 F_1 - (\xi + \vartheta) + \frac{\xi F_1(\varphi_1 \mu + \varphi_3(\kappa + \sigma U_1))}{\theta(\kappa + \sigma U_1)} \right) Q_1 = 0,$$

$$\left( \varphi_3 F_1 - \frac{F_1(\varphi_1 \mu + \varphi_3(\kappa + \sigma U_1))}{\kappa + \sigma U_1} + \frac{\mu \varphi_1 F_1}{\kappa + \sigma U_1} \right) G_1 = 0.$$

Therefore, Equation (22) will be represented in the following manner:

$$\frac{d\Theta_1}{dt} = \left( 1 - \frac{F_1}{F} \right)(\rho F_1 - \rho F) + (\varphi_1 F_1 K_1 + \varphi_2 F_1 Q_1 + \varphi_3 F_1 G_1)\left( 1 - \frac{F_1}{F} \right) + \varphi_1 F_1 K$$

$$- \varphi_1 F_1 K_1 \frac{FKQ_1}{F_1 K_1 Q} - \varphi_2 F_1 Q_1 \frac{F}{F_1} - \varphi_3 F_1 G_1 \frac{FGQ_1}{F_1 G_1 Q} + \varphi_1 F_1 K_1$$

$$+ \varphi_2 F_1 Q_1 + \varphi_3 F_1 G_1 - \frac{\xi F_1(\varphi_1 \mu + \varphi_3(\kappa + \sigma U_1)) Q_1}{\theta(\kappa + \sigma U_1)} \frac{QG_1}{Q_1 G} + \varphi_1 F_1 K_1$$

$$+ \varphi_3 F_1 G_1 - \frac{\mu \varphi_1 F_1 G_1}{\kappa + \sigma U_1} \frac{GK_1}{G_1 K} - \frac{\kappa \varphi_1 F_1}{\kappa + \sigma U_1}(K - K_1) - \frac{\sigma \varphi_1 F_1}{\kappa + \sigma U_1} U(K - K_1)$$

$$+ \frac{\sigma \varphi_1 F_1}{\kappa + \sigma U_1} U_1(K - K_1) - \frac{\sigma \varphi_1 F_1}{\kappa + \sigma U_1} U_1(K - K_1)$$

$$+ \frac{\sigma \varphi_1 F_1}{(\kappa + \sigma U_1)(\phi - \varepsilon U_1)}(U - U_1)(\phi K - \gamma U - \varepsilon UK - \phi K_1 + \gamma U_1 + \varepsilon U_1 K_1 + \varepsilon U_1 K - \varepsilon U_1 K)$$

$$= -\frac{\rho(F - F_1)^2}{F} + (\varphi_1 F_1 K_1 + \varphi_2 F_1 Q_1 + \varphi_3 F_1 G_1)\left( 2 - \frac{F_1}{F} \right) + \varphi_1 F_1 K - \varphi_1 F_1 K_1 \frac{FKQ_1}{F_1 K_1 Q}$$

$$- \varphi_2 F_1 Q_1 \frac{F}{F_1} - \varphi_3 F_1 G_1 \frac{FGQ_1}{F_1 G_1 Q} - (\varphi_1 F_1 K_1 + \varphi_3 F_1 G_1)\frac{QG_1}{Q_1 G}$$

$$+ \varphi_1 F_1 K_1 + \varphi_3 F_1 G_1 - \varphi_1 F_1 K_1 \frac{GK_1}{G_1 K} - \frac{\varphi_1 F_1(\kappa + \sigma U_1)}{\kappa + \sigma U_1}(K - K_1)$$

$$- \frac{\sigma \varphi_1 F_1}{\kappa + \sigma U_1}(U - U_1)(K - K_1) + \frac{\sigma \varphi_1 F_1(\phi - \varepsilon U_1)}{(\kappa + \sigma U_1)(\phi - \varepsilon U_1)}(U - U_1)(K - K_1)$$

$$- \frac{\sigma \varphi_1 F_1(\gamma + \varepsilon K)}{(\kappa + \sigma U_1)(\phi - \varepsilon U_1)}(U - U_1)^2$$

$$= -\frac{\rho(F - F_1)^2}{F} + (\varphi_1 F_1 K_1 + \varphi_2 F_1 Q_1 + \varphi_3 F_1 G_1)\left( 2 - \frac{F_1}{F} \right) + \varphi_1 F_1 K - \varphi_1 F_1 K_1 \frac{FKQ_1}{F_1 K_1 Q}$$

$$- \varphi_2 F_1 Q_1 \frac{F}{F_1} - \varphi_3 F_1 G_1 \frac{FGQ_1}{F_1 G_1 Q} - (\varphi_1 F_1 K_1 + \varphi_3 F_1 G_1)\frac{QG_1}{Q_1 G}$$

$$+ \varphi_1 F_1 K_1 + \varphi_3 F_1 G_1 - \varphi_1 F_1 K_1 \frac{GK_1}{G_1 K} - \varphi_1 F_1 K_1\left( \frac{K}{K_1} - 1 \right)$$

$$- \frac{\sigma \varphi_1 F_1(\gamma + \varepsilon K)}{(\kappa + \sigma U_1)(\phi - \varepsilon U_1)}(U - U_1)^2$$

$$
\begin{aligned}
= & -\frac{\rho(F-F_1)^2}{F} + (\varphi_1 F_1 K_1 + \varphi_2 F_1 Q_1 + \varphi_3 F_1 G_1)\left(2 - \frac{F_1}{F}\right) + \varphi_1 F_1 K - \varphi_1 F_1 K_1 \frac{FKQ_1}{F_1 K_1 Q} \\
& - \varphi_2 F_1 Q_1 \frac{F}{F_1} - \varphi_3 F_1 G_1 \frac{FGQ_1}{F_1 G_1 Q} - (\varphi_1 F_1 K_1 + \varphi_3 F_1 G_1)\frac{QG_1}{Q_1 G} \\
& + 2\varphi_1 F_1 K_1 + \varphi_3 F_1 G_1 - \varphi_1 F_1 K_1 \frac{GK_1}{G_1 K} - \varphi_1 F_1 K - \frac{\sigma \varphi_1 F_1 (\gamma + \varepsilon K)}{(\kappa + \sigma U_1)(\phi - \varepsilon U_1)}(U - U_1)^2 \\
= & -\frac{\rho(F-F_1)^2}{F} + \varphi_1 F_1 K_1 \left(4 - \frac{F_1}{F} - \frac{FKQ_1}{F_1 K_1 Q} - \frac{QG_1}{Q_1 G} - \frac{GK_1}{G_1 K}\right) + \varphi_2 F_1 Q_1 \left(2 - \frac{F_1}{F} - \frac{F}{F_1}\right) \\
& + \varphi_3 F_1 G_1 \left(3 - \frac{F_1}{F} - \frac{FGQ_1}{F_1 G_1 Q} - \frac{QG_1}{Q_1 G}\right) - \frac{\sigma \varphi_1 F_1 (\gamma + \varepsilon K)}{(\kappa + \sigma U_1)(\phi - \varepsilon U_1)}(U - U_1)^2.
\end{aligned}
$$

Thus

$$
\begin{aligned}
\frac{d\Theta_1}{dt} = & -\frac{(\rho + \varphi_2 Q_1)(F - F_1)^2}{F} + \varphi_1 F_1 K_1 \left(4 - \frac{F_1}{F} - \frac{FKQ_1}{F_1 K_1 Q} - \frac{QG_1}{Q_1 G} - \frac{GK_1}{G_1 K}\right) \\
& + \varphi_3 F_1 G_1 \left(3 - \frac{F_1}{F} - \frac{FGQ_1}{F_1 G_1 Q} - \frac{QG_1}{Q_1 G}\right) - \frac{\sigma \varphi_1 F_1 (\gamma + \varepsilon K)}{(\kappa + \sigma U_1)(\phi - \varepsilon U_1)}(U - U_1)^2.
\end{aligned}
$$

The AM-GM inequality tells us that

$$
4 \leq \frac{F_1}{F} + \frac{FKQ_1}{F_1 K_1 Q} + \frac{QG_1}{Q_1 G} + \frac{GK_1}{G_1 K},
$$
$$
3 \leq \frac{F_1}{F} + \frac{FGQ_1}{F_1 G_1 Q} + \frac{QG_1}{Q_1 G}.
$$

Hence, if $\Re_0 > 1$, then $\frac{d\Theta_1}{dt} \leq 0$ for all $F, Q, G, K, U > 0$. Also, $\frac{d\Theta_1}{dt} = 0$ when $F = F_1$, $Q = Q_1, G = G_1, K = K_1$ and $U = U_1$. Let $\Phi_1'$ be the largest invariant subset of $\Phi_1 = \left\{(F, Q, G, K, U) : \frac{d\Theta_1}{dt} = 0\right\}$. Therefore, $\Phi_1' = \{\Xi Q_1\}$. By applying L.I.P, we can conclude that if $\Re_0 > 1$, then the equilibrium $\Xi Q_1$ is G.A.S [42]. $\quad\square$

## 3. Modeling Hiv-1 with Distributed Delays

### 3.1. Description of the System

In the subsequent model, we incorporate the distributed time-delays in system (5) by representing them as delay differential equations (DDEs):

$$
\begin{cases}
\frac{dF(t)}{dt} = \psi - \rho F(t) - \varphi_1 F(t)K(t) - \varphi_2 F(t)Q(t) - \varphi_3 F(t)G(t), \\
\frac{dQ(t)}{dt} = \int_0^{\varrho_1} T_1(\lambda)e^{-\alpha_1 \lambda} F(t-\lambda)(\varphi_1 K(t-\lambda) + \varphi_2 Q(t-\lambda) \\
\qquad\qquad + \varphi_3 G(t-\lambda))d\lambda - (\xi + \vartheta)Q(t), \\
\frac{dG(t)}{dt} = \xi \int_0^{\varrho_2} T_2(\lambda)e^{-\alpha_2 \lambda} Q(t-\lambda)d\lambda - \theta G(t), \\
\frac{dK(t)}{dt} = \mu \int_0^{\varrho_3} T_3(\lambda)e^{-\alpha_3 \lambda} G(t-\lambda)d\lambda - \kappa K(t) - \sigma U(t)K(t), \\
\frac{dU(t)}{dt} = \phi K(t) - \gamma U(t) - \varepsilon U(t)K(t).
\end{cases}
\tag{23}
$$

Here, system (23) includes the following assumptions:

- Healthy cells, which are contacted by HIV-1 particles or infected cells at time $t$, become (HIV-1)-latently infected cells, $\lambda$ time units later. The recruitment of (HIV-1)-latently infected cells at time $t$ is given by the number of cells that were newly contacted at time $t - \lambda$ and are still alive at time $t$. Here, $\alpha_1$ is assumed to be a constant death rate for contacted cells. Thus, the probability of surviving the time period from $t - \lambda$ to $t$ is $T_1(\lambda)e^{-\alpha_1 \lambda}$.
- (HIV-1)-latently infected cells, become (HIV-1)-actively infected cells, $\lambda$ time units later. The recruitment of (HIV-1)-actively infected cells at time $t$ is given by the number of cells that were newly being (HIV-1)-latently infected cells at time $t - \lambda$ and are still

alive at time $t$. Here, $\alpha_2$ is assumed to be a constant death rate for (HIV-1)-latently infected cells. Thus, the probability of surviving the time period from $t - \lambda$ to $t$ is $T_2(\lambda)e^{-\alpha_2\lambda}$.

- (HIV-1)-actively infected cells, produce new mature HIV-1 particles, $\lambda$ time units later. The recruitment of HIV-1 particles at time $t$ is given by the number of cells that were newly being (HIV-1)-actively infected cells at time $t - \lambda$ and are still alive at time $t$. Here, $\alpha_3$ is assumed to be a constant death rate for (HIV-1)-actively infected cells. Thus, the probability of surviving the time period from $t - \lambda$ to $t$ is $T_3(\lambda)e^{-\alpha_3\lambda}$.

The delay parameter $\lambda$ is obtained from a probability distribution function $T_i(\lambda)$ over the interval $[0, \varrho_i]$, where $\varrho_i$ is the upper limit of the delay period. The functions $T_i(\lambda)$, $i = 1, 2, 3$, satisfy the following conditions:

$$T_i(\lambda) > 0, \quad \int_0^{\varrho_i} T_i(\lambda)d\lambda = 1, \quad \text{and} \quad \int_0^{\varrho_i} T_i(\lambda)e^{-\beta\lambda}d\lambda < \infty, \quad \text{where} \quad \beta > 0.$$

Let $\bar{\Lambda}_i(\lambda) = T_i(\lambda)e^{-\alpha_i\lambda}$ and $\Lambda_i = \int_0^{\varrho_i} \bar{\Lambda}_i(\lambda)d\lambda$, $i = 1, 2, 3$. Therefore, $0 < \Lambda_i \le 1$, $i = 1, 2, 3$. The initial conditions of system (23) are:

$$\begin{cases} F(r) = a_1(r), & Q(r) = a_2(r), & G(r) = a_3(r), & K(r) = a_4(r), & U(r) = a_5(r), \\ a_j(r) \ge 0, & j = 1, 2, ..., 5, & r \in [-\varrho, 0], & \varrho = \max\{\varrho_1, \varrho_2, \varrho_3\}, \end{cases} \tag{24}$$

where $a_j(r) \in C([-\varrho, 0], \mathbb{R}_{\ge 0})$, $j = 1, 2, ..., 5$ and $C = C([-\varrho, 0], \mathbb{R}_{\ge 0})$ is the Banach space of continuous functions with norm $\|a_j\| = \sup\limits_{-\varrho \le \zeta \le 0} |a_j(\zeta)|$ for all $a_j \in C$. Therefore, system (23) with initial conditions (24) has a unique solution [42,43]. The biological interpretations of all the other parameters and variables are the same as those explained in Section 2.

*3.2. Main Basic Properties*

3.2.1. Maintaining Non-Negativity and Ultimate Boundedness in the Solutions

**Lemma 3.** *For system (23), along with the initial conditions (24), there exists a positively invariant compact set denoted by $\hat{\Omega}$, defined as follows:*

$$\hat{\Omega} = \{(F, Q, G, K, U) \in C_{\ge 0}^5 : \|F(t)\| \le \hat{\tau}_1, \|Q(t)\| \le \hat{\tau}_1, \|G(t)\| \le \hat{\tau}_2, \|K(t)\| \le \hat{\tau}_3, \|U(t)\| \le \hat{\tau}_4\}.$$

**Proof.** Since $\left.\dfrac{dF}{dt}\right|_{F=0} = \psi > 0$, we can conclude that $F(t) > 0$ for all $t \ge 0$. Additionally, the other equations in system (23) can be expressed as:

$$\frac{dQ(t)}{dt} + (\xi + \vartheta)Q(t) = \int_0^{\varrho_1} \bar{\Lambda}_1(\lambda)F(t - \lambda)(\varphi_1 K(t - \lambda) + \varphi_2 Q(t - \lambda) + \varphi_3 G(t - \lambda))d\lambda$$

$$\implies Q(t) = a_2(0)e^{-(\xi+\vartheta)t} + \int_0^t e^{-(\xi+\vartheta)(t-\varkappa)} \int_0^{\varrho_1} \bar{\Lambda}_1(\lambda)F(\varkappa - \lambda)(\varphi_1 K(\varkappa - \lambda)$$
$$+ \varphi_2 Q(\varkappa - \lambda) + \varphi_3 G(\varkappa - \lambda))d\lambda d\varkappa \ge 0.$$

$$\frac{dG(t)}{dt} + \theta G(t) = \xi \int_0^{\varrho_2} \bar{\Lambda}_2(\lambda)Q(t - \lambda)d\lambda$$

$$\implies G(t) = a_3(0)e^{-\theta t} + \xi \int_0^t e^{-\theta(t-\varkappa)} \int_0^{\varrho_2} \bar{\Lambda}_2(\lambda)Q(\varkappa - \lambda)d\lambda d\varkappa \ge 0.$$

$$\frac{dK(t)}{dt} + (\kappa + \sigma U(t))K(t) = \mu \int_0^{\varrho_3} \bar{\Lambda}_3(\lambda)G(t - \lambda)d\lambda$$

$$\implies K(t) = a_4(0)e^{-\int_0^t (\kappa+\sigma U(u))du} + \mu \int_0^t e^{-\int_\varkappa^t (\kappa+\sigma U(u))du} \int_0^{\varrho_3} \bar{\Lambda}_3(\lambda)G(\varkappa - \lambda)d\lambda d\varkappa \ge 0.$$

$$\frac{dU(t)}{dt} + (\gamma + \varepsilon K(t))U(t) = \phi K(t)$$

$$\implies U(t) = a_5(0)e^{-\int_0^t (\gamma+\varepsilon K(u))du} + \phi \int_0^t e^{-\int_\varkappa^t (\gamma+\varepsilon K(u))du} K(\lambda)d\lambda \ge 0,$$

for all $t \in [0, \varrho]$.

By a recursive argument, it can be established that $F(t)$, $Q(t)$, $G(t)$, $K(t)$, and $U(t)$ are nonnegative for all $t \geq 0$. Consequently, the solutions of system (23) satisfy $(F(t), Q(t), G(t), K(t), U(t)) \in \mathbb{R}^5_{\geq 0}$ for all $t \geq 0$.

Using the first equation of system (23), we can deduce that $\limsup_{t \to \infty} F(t) \leq \frac{\psi}{\rho}$. Next, we define

$$Y_1(t) = \int_0^{\varrho_1} \bar{\Lambda}_1(\lambda) F(t - \lambda) d\lambda + Q(t).$$

Then

$$
\begin{aligned}
\frac{dY_1(t)}{dt} &= \int_0^{\varrho_1} \bar{\Lambda}_1(\lambda) \frac{dF(t - \lambda)}{dt} d\lambda + \frac{dQ(t)}{dt} \\
&= \int_0^{\varrho_1} \bar{\Lambda}_1(\lambda)(\psi - \rho F(t - \lambda)) d\lambda - (\xi + \vartheta) Q(t) \\
&= \psi \Lambda_1 - \rho \int_0^{\varrho_1} \bar{\Lambda}_1(\lambda) F(t - \lambda) d\lambda - (\xi + \vartheta) Q(t) \\
&\leq \psi - \eta_1 \left( \int_0^{\varrho_1} \bar{\Lambda}_1(\lambda) F(t - \lambda) d\lambda + Q(t) \right) = \psi - \eta_1 Y_1(t),
\end{aligned}
$$

where $\eta_1 = \min\{\rho, \xi + \vartheta\}$. Hence, $\limsup_{t \to \infty} Y_1(t) \leq \frac{\psi}{\eta_1} = \hat{\tau}_1$. Since $\int_0^{\varrho_1} \bar{\Lambda}_1(\lambda) F(t - \lambda) d\lambda$ and $Q(t)$ are nonnegative, then $\limsup_{t \to \infty} Q(t) \leq \hat{\tau}_1$. Additionally, from the third equation of system (23), we get

$$\frac{dG(t)}{dt} = \xi \int_0^{\varrho_2} \bar{\Lambda}_2(\lambda) Q(t - \lambda) d\lambda - \theta G(t) \leq \xi \Lambda_2 \hat{\tau}_1 - \theta G(t) \leq \xi \hat{\tau}_1 - \theta G(t).$$

Therefore, $\limsup_{t \to \infty} G(t) \leq \frac{\xi \hat{\tau}_1}{\theta} = \hat{\tau}_2$. Finally, we let

$$Y_2(t) = K(t) + \frac{\kappa}{2\phi} U(t).$$

This yields

$$
\begin{aligned}
\frac{dY_2(t)}{dt} &= \frac{dK(t)}{dt} + \frac{\kappa}{2\phi} \frac{dU(t)}{dt} \\
&= \mu \int_0^{\varrho_3} \bar{\Lambda}_3(\lambda) G(t - \lambda) d\lambda - \kappa K(t) - \sigma U(t) K(t) \\
&\quad + \frac{\kappa}{2\phi}(\phi K(t) - \gamma U(t) - \varepsilon U(t) K(t)) \\
&= \mu \int_0^{\varrho_3} \bar{\Lambda}_3(\lambda) G(t - \lambda) d\lambda - \frac{\kappa}{2} K(t) - \frac{\kappa \gamma}{2\phi} U(t) - (\sigma + \frac{\kappa \varepsilon}{2\phi}) U(t) K(t) \\
&\leq \mu \int_0^{\varrho_3} \bar{\Lambda}_3(\lambda) G(t - \lambda) d\lambda - \frac{\kappa}{2} K(t) - \frac{\kappa \gamma}{2\phi} U(t) \\
&\leq \mu \hat{\tau}_2 - \eta_2 (K(t) + \frac{\kappa}{2\phi} U(t)) = \mu \hat{\tau}_2 - \eta_2 Y_2(t),
\end{aligned}
$$

where $\eta_2 = \min\{\frac{\kappa}{2}, \gamma\}$. Hence, $\limsup_{t \to \infty} Y_2(t) \leq \frac{\mu \hat{\tau}_2}{\eta_2} = \hat{\tau}_3$. We have $K(t)$ and $U(t)$ are nonnegative, this guarantees that $\limsup_{t \to \infty} K(t) \leq \hat{\tau}_3$, and $\limsup_{t \to \infty} U(t) \leq \frac{2\phi \hat{\tau}_3}{\kappa} = \hat{\tau}_4$. We conclude that $F(t)$, $Q(t)$, $G(t)$, $K(t)$ and $U(t)$ are ultimately bounded. Hence, the compact set $\hat{\Omega}$ remains positively invariant under the dynamics of system (23). □

3.2.2. Analysis of Reproductive Numbers and Equilibrium Points

**Lemma 4.** *There exists a basic reproductive number $\tilde{\Re}_0 = \frac{\Lambda_1 \tilde{F}_0(\Lambda_2\xi(\Lambda_3\varphi_1\mu+\varphi_3\kappa)+\varphi_2\kappa\theta)}{(\xi+\vartheta)\kappa\theta}$ for system (23) such that*

*(i)    the system always has an infection-free equilibrium point $\Xi\tilde{Q}_0$, and*

*(ii)   if $\tilde{\Re}_0 > 1$, the system also has an infected equilibrium point $\Xi\tilde{Q}_1$.*

**Proof.** It is evident that system (23) possesses an infection-free equilibrium denoted as $\Xi\tilde{Q}_0 = (\tilde{F}_0, 0, 0, 0, 0)$, where $\tilde{F}_0 = \frac{\psi}{\rho}$. The nonlinear terms responsible for new infections, denoted as $\hat{\Gamma}_2$, and the outflow term $\hat{\Delta}_2$, are represented by:

$$\hat{\Gamma}_2 = \begin{pmatrix} \Lambda_1(\varphi_1 FK + \varphi_2 FQ + \varphi_3 FG) \\ 0 \\ 0 \end{pmatrix}, \quad \hat{\Delta}_2 = \begin{pmatrix} (\xi+\vartheta)Q \\ -\xi\Lambda_2 Q + \theta\, G \\ -\mu\Lambda_3 G + \kappa K + \sigma UK \end{pmatrix}.$$

We calculate the derivatives of $\hat{\Gamma}_2$ and $\hat{\Delta}_2$ at the equilibrium $\Xi\tilde{Q}_0$, resulting in the following matrices:

$$\Gamma_2 = \begin{pmatrix} \Lambda_1\varphi_2\tilde{F}_0 & \Lambda_1\varphi_3\tilde{F}_0 & \Lambda_1\varphi_1\tilde{F}_0 \\ 0 & 0 & 0 \\ 0 & 0 & 0 \end{pmatrix}, \quad \Delta_2 = \begin{pmatrix} \xi+\vartheta & 0 & 0 \\ -\xi\Lambda_2 & \theta & 0 \\ 0 & -\mu\Lambda_3 & \kappa \end{pmatrix}.$$

The next-generation matrix takes the following form:

$$\Gamma_2\Delta_2^{-1} = \begin{pmatrix} \frac{\Lambda_1\tilde{F}_0(\Lambda_2\xi(\Lambda_3\varphi_1\mu+\varphi_3\kappa)+\varphi_2\kappa\theta)}{(\xi+\vartheta)\kappa\theta} & \frac{\Lambda_1\tilde{F}_0(\Lambda_3\varphi_1\mu+\varphi_3\kappa)}{\kappa\theta} & \frac{\Lambda_1\tilde{F}_0\varphi_1}{\kappa} \\ 0 & 0 & 0 \\ 0 & 0 & 0 \end{pmatrix}.$$

The basic reproductive number $\tilde{\Re}_0$ is given as:

$$\tilde{\Re}_0 = \frac{\Lambda_1\tilde{F}_0(\Lambda_2\xi(\Lambda_3\varphi_1\mu+\varphi_3\kappa)+\varphi_2\kappa\theta)}{(\xi+\vartheta)\kappa\theta} = \tilde{\Re}_{01} + \tilde{\Re}_{02} + \tilde{\Re}_{03}, \tag{25}$$

where

$$\tilde{\Re}_{01} = \frac{\Lambda_1\Lambda_2\Lambda_3\tilde{F}_0\mu\xi\varphi_1}{\kappa\theta(\xi+\vartheta)}, \quad \tilde{\Re}_{02} = \frac{\Lambda_1\tilde{F}_0\varphi_2}{\xi+\vartheta}, \quad \tilde{\Re}_{03} = \frac{\Lambda_1\Lambda_2\tilde{F}_0\xi\varphi_3}{\theta(\xi+\vartheta)}.$$

The parameters $\tilde{\Re}_{0i}$, $i = 1, 2, 3$ have the same biological meaning of the parameters $\Re_{0i}$, $i = 1, 2, 3$ that explained in Section 2. To find any additional equilibrium to $\Xi\tilde{Q}_0$, we let $(F, Q, G, K, U)$ be any equilibrium satisfying

$$0 = \psi - \rho F - \varphi_1 FK - \varphi_2 FQ - \varphi_3 FG, \tag{26}$$

$$0 = \Lambda_1(\varphi_1 FK + \varphi_2 FQ + \varphi_3 FG) - (\xi+\vartheta)Q, \tag{27}$$

$$0 = \xi\Lambda_2 Q - \theta G, \tag{28}$$

$$0 = \mu\Lambda_3 G - \kappa K - \sigma UK, \tag{29}$$

$$0 = \phi K - \gamma U - \varepsilon UK. \tag{30}$$

From Equations (28) and (30), we get

$$Q = \frac{\theta G}{\xi\Lambda_2}, \quad U = \frac{\phi K}{\gamma+\varepsilon K}. \tag{31}$$

Upon substitution of Equation (31) into Equation (29), the result is as follows:

$$G = \frac{\gamma\kappa K + (\sigma\phi + \kappa\varepsilon)K^2}{\mu\Lambda_3(\gamma + \varepsilon K)}. \tag{32}$$

Upon substitution of Equation (32) into Equation (31), the result is as follows:

$$Q = \frac{\theta(\gamma\kappa K + (\sigma\phi + \kappa\varepsilon)K^2)}{\mu\xi\Lambda_2\Lambda_3(\gamma + \varepsilon K)}. \tag{33}$$

From Equations (26) and (27), we get

$$\psi - \rho F = \frac{(\xi + \vartheta)Q}{\Lambda_1}. \tag{34}$$

Upon substitution of Equation (33) into Equation (34), the result is as follows:

$$F = \frac{1}{\rho}\left(\psi - \frac{\theta(\xi + \vartheta)(\gamma\kappa K + (\sigma\phi + \kappa\varepsilon)K^2)}{\mu\xi\Lambda_1\Lambda_2\Lambda_3(\gamma + \varepsilon K)}\right). \tag{35}$$

Upon substitution of Equations (32), (33) and (35) into Equation (27), the result is as follows:

$$\frac{K}{\rho\mu^2\xi^2\Lambda_2^2\Lambda_3^2(\gamma + \varepsilon K)^2}\left(\tilde{A}K^3 + \tilde{B}K^2 + \tilde{C}K + \tilde{D}\right) = 0, \tag{36}$$

where

$$\begin{aligned}
\tilde{A} &= \theta(\xi + \vartheta)(\sigma\phi + \varepsilon\kappa)(\theta\varphi_2(\sigma\phi + \varepsilon\kappa) + \xi\Lambda_2(\varepsilon\mu\Lambda_3\varphi_1 + \varphi_3(\sigma\phi + \varepsilon\kappa))), \\
\tilde{B} &= \varepsilon\mu\rho\xi\theta\Lambda_2\Lambda_3(\xi + \vartheta)(\sigma\phi + \varepsilon\kappa) + \theta(\xi + \vartheta)(\sigma\phi + \varepsilon\kappa)(\gamma\kappa\theta\varphi_2 \\
&\quad + \gamma\xi\Lambda_2(\mu\Lambda_3\varphi_1 + \kappa\varphi_3)) + (\gamma\kappa\theta(\xi + \vartheta) - \psi\varepsilon\mu\xi\Lambda_1\Lambda_2\Lambda_3) \\
&\quad (\theta\varphi_2(\sigma\phi + \varepsilon\kappa) + \xi\Lambda_2(\varepsilon\mu\Lambda_3\varphi_1 + \varphi_3(\sigma\phi + \varepsilon\kappa))), \\
\tilde{C} &= \gamma\Big(\gamma\varphi_2\kappa^2\theta^2(\xi + \vartheta) - \psi\mu\Lambda_1\Lambda_3\xi^2\Lambda_2^2(2\varepsilon\mu\Lambda_3\varphi_1 + \varphi_3(\sigma\phi + 2\varepsilon\kappa)) \\
&\quad + \xi\theta\Lambda_2(\mu\Lambda_3(\gamma\kappa\varphi_1(\xi + \vartheta) + (\sigma\phi + 2\varepsilon\kappa)(\rho(\xi + \vartheta) - \psi\Lambda_1\varphi_2)) \\
&\quad + \gamma\varphi_3\kappa^2(\xi + \vartheta))\Big), \\
\tilde{D} &= \mu\rho\kappa\xi\theta\Lambda_2\Lambda_3\gamma^2(\xi + \vartheta)(1 - \tilde{\mathfrak{R}}_0),
\end{aligned}$$

where $\tilde{\mathfrak{R}}_0$ is defined by Equation (25). From Equation (36), we have

1.  If $K = 0$, then from Equations (31)–(33) and (35) the infection-free equilibrium $\Xi\tilde{Q}_0$ is obtained.
2.  If $K \neq 0$, we have the equation $\tilde{A}K^3 + \tilde{B}K^2 + \tilde{C}K + \tilde{D} = 0$. In this scenario, let us introduce a function $\bar{\Psi}(K)$ defined on $[0, \infty)$ as:

$$\bar{\Psi}(K) = \tilde{A}K^3 + \tilde{B}K^2 + \tilde{C}K + \tilde{D}.$$

We have

$$\begin{aligned}
\bar{\Psi}(0) &= \mu\rho\kappa\xi\theta\Lambda_2\Lambda_3\gamma^2(\xi + \vartheta)(1 - \tilde{\mathfrak{R}}_0) < 0, \text{ if } \tilde{\mathfrak{R}}_0 > 1, \\
\lim_{K \to \infty} \bar{\Psi}(K) &= \infty.
\end{aligned}$$

As a result, the function $\bar{\Psi}$ possesses a positive real root denoted as $\tilde{K}_1 > 0$. Consequently, by substituting Equations (32) and (33) into Equation (26), we obtain:

$$\tilde{F}_1 = \frac{\psi}{\rho + \varphi_1\tilde{K}_1 + \varphi_2\tilde{Q}_1 + \varphi_3\tilde{G}_1},$$

where

$$\tilde{Q}_1 = \frac{\theta(\gamma\kappa\tilde{K}_1 + (\sigma\phi + \kappa\varepsilon)\tilde{K}_1^2)}{\mu\xi\Lambda_2\Lambda_3(\gamma + \varepsilon\tilde{K}_1)}, \quad \tilde{G}_1 = \frac{\gamma\kappa\tilde{K}_1 + (\sigma\phi + \kappa\varepsilon)\tilde{K}_1^2}{\mu\Lambda_3(\gamma + \varepsilon\tilde{K}_1)}, \quad \tilde{U}_1 = \frac{\phi\tilde{K}_1}{\gamma + \varepsilon\tilde{K}_1}.$$

The existence of the infected equilibrium $\Xi\tilde{Q}_1 = (\tilde{F}_1, \tilde{Q}_1, \tilde{G}_1, \tilde{K}_1, \tilde{U}_1)$ is contingent upon $\tilde{\Re}_0$ being greater than 1. $\square$

### 3.2.3. The Analysis of the Stability of the Equilibria $\Xi\tilde{Q}_0$ and $\Xi\tilde{Q}_1$

In the upcoming theorems, we will explore the concept of global asymptotic stability concerning the equilibrium points. In order to simplify the following discussion, we denote $(F(t - \lambda), Q(t - \lambda), G(t - \lambda), K(t - \lambda))$ by $(F_\lambda, Q_\lambda, G_\lambda, K_\lambda)$.

**Theorem 4.** *For system* (23), *if* $\tilde{\Re}_0 < 1$, *then* $\Xi\tilde{Q}_0$ *is G.A.S, and it becomes unstable when* $\tilde{\Re}_0 > 1$.

**Proof.** We introduce a Lyapunov function as follows:

$$\tilde{\Theta}_0 = \tilde{F}_0 H\left(\frac{F}{\tilde{F}_0}\right) + \frac{1}{\Lambda_1}Q + \frac{(\xi + \vartheta)(1 - \tilde{\Re}_{02})}{\xi\Lambda_1\Lambda_2}G + \frac{\theta(\xi + \vartheta)(1 - (\tilde{\Re}_{02} + \tilde{\Re}_{03}))}{\xi\mu\Lambda_1\Lambda_2\Lambda_3}K$$

$$+ \frac{\theta\kappa(\xi + \vartheta)(1 - \tilde{\Re}_0)}{\phi\xi\mu\Lambda_1\Lambda_2\Lambda_3}U + \frac{1}{\Lambda_1}\int_0^{\varrho_1} \bar{\Lambda}_1(\lambda)\int_{t-\lambda}^t F(\varkappa)(\varphi_1 K(\varkappa) + \varphi_2 Q(\varkappa) + \varphi_3 G(\varkappa))d\varkappa d\lambda$$

$$+ \frac{(\xi + \vartheta)(1 - \tilde{\Re}_{02})}{\Lambda_1\Lambda_2}\int_0^{\varrho_2} \bar{\Lambda}_2(\lambda)\int_{t-\lambda}^t Q(\varkappa)d\varkappa d\lambda$$

$$+ \frac{\theta(\xi + \vartheta)(1 - (\tilde{\Re}_{02} + \tilde{\Re}_{03}))}{\xi\Lambda_1\Lambda_2\Lambda_3}\int_0^{\varrho_3} \bar{\Lambda}_3(\lambda)\int_{t-\lambda}^t G(\varkappa)d\varkappa d\lambda.$$

Clearly, $\tilde{\Theta}_0(F, Q, G, K, U) > 0$ for all $F, Q, G, K, U > 0$, and $\tilde{\Theta}_0(\tilde{F}_0, 0, 0, 0, 0) = 0$. Further, $\frac{d\tilde{\Theta}_0}{dt}$ is given by:

$$\frac{d\tilde{\Theta}_0}{dt} = \left(1 - \frac{\tilde{F}_0}{F}\right)\frac{dF}{dt} + \frac{1}{\Lambda_1}\frac{dQ}{dt} + \frac{(\xi + \vartheta)(1 - \tilde{\Re}_{02})}{\xi\Lambda_1\Lambda_2}\frac{dG}{dt} + \frac{\theta(\xi + \vartheta)(1 - (\tilde{\Re}_{02} + \tilde{\Re}_{03}))}{\xi\mu\Lambda_1\Lambda_2\Lambda_3}\frac{dK}{dt}$$

$$+ \frac{\theta\kappa(\xi + \vartheta)(1 - \tilde{\Re}_0)}{\phi\xi\mu\Lambda_1\Lambda_2\Lambda_3}\frac{dU}{dt} + \varphi_1 FK + \varphi_2 FQ + \varphi_3 FG$$

$$- \frac{1}{\Lambda_1}\int_0^{\varrho_1} \bar{\Lambda}_1(\lambda)F_\lambda(\varphi_1 K_\lambda + \varphi_2 Q_\lambda + \varphi_3 G_\lambda)d\lambda$$

$$+ \frac{(\xi + \vartheta)(1 - \tilde{\Re}_{02})}{\Lambda_1}Q - \frac{(\xi + \vartheta)(1 - \tilde{\Re}_{02})}{\Lambda_1\Lambda_2}\int_0^{\varrho_2} \bar{\Lambda}_2(\lambda)Q_\lambda d\lambda$$

$$+ \frac{\theta(\xi + \vartheta)(1 - (\tilde{\Re}_{02} + \tilde{\Re}_{03}))}{\xi\Lambda_1\Lambda_2}G - \frac{\theta(\xi + \vartheta)(1 - (\tilde{\Re}_{02} + \tilde{\Re}_{03}))}{\xi\Lambda_1\Lambda_2\Lambda_3}\int_0^{\varrho_3} \bar{\Lambda}_3(\lambda)G_\lambda d\lambda.$$

This implies that

$$\frac{d\tilde{\Theta}_0}{dt} = \left(1 - \frac{\tilde{F}_0}{F}\right)(\psi - \rho F - \varphi_1 FK - \varphi_2 FQ - \varphi_3 FG) + \frac{1}{\Lambda_1}\left(\int_0^{\varrho_1} \bar{\Lambda}_1(\lambda)F_\lambda(\varphi_1 K_\lambda\right.$$

$$+ \varphi_2 Q_\lambda + \varphi_3 G_\lambda)d\lambda - (\xi + \vartheta)Q\Bigg) + \frac{(\xi + \vartheta)(1 - \tilde{\Re}_{02})}{\xi\Lambda_1\Lambda_2}\left(\xi\int_0^{\varrho_2} \bar{\Lambda}_2(\lambda)Q_\lambda d\lambda - \theta G\right)$$

$$+ \frac{\theta(\xi + \vartheta)(1 - (\tilde{\Re}_{02} + \tilde{\Re}_{03}))}{\xi\mu\Lambda_1\Lambda_2\Lambda_3}\left(\mu\int_0^{\varrho_3} \bar{\Lambda}_3(\lambda)G_\lambda d\lambda - \kappa K - \sigma UK\right)$$

$$+ \frac{\theta\kappa(\xi + \vartheta)(1 - \tilde{\Re}_0)}{\phi\xi\mu\Lambda_1\Lambda_2\Lambda_3}(\phi K - \gamma U - \varepsilon UK) + \varphi_1 FK + \varphi_2 FQ + \varphi_3 FG$$

$$- \frac{1}{\Lambda_1}\int_0^{\varrho_1} \bar{\Lambda}_1(\lambda)F_\lambda(\varphi_1 K_\lambda + \varphi_2 Q_\lambda + \varphi_3 G_\lambda)d\lambda + \frac{(\xi + \vartheta)(1 - \tilde{\Re}_{02})}{\Lambda_1}Q$$

$$- \frac{(\xi + \vartheta)(1 - \tilde{\mathfrak{R}}_{02})}{\Lambda_1 \Lambda_2} \int_0^{\varrho_2} \bar{\Lambda}_2(\lambda) Q_\lambda d\lambda + \frac{\theta(\xi + \vartheta)(1 - (\tilde{\mathfrak{R}}_{02} + \tilde{\mathfrak{R}}_{03}))}{\xi \Lambda_1 \Lambda_2} G$$

$$- \frac{\theta(\xi + \vartheta)(1 - (\tilde{\mathfrak{R}}_{02} + \tilde{\mathfrak{R}}_{03}))}{\xi \Lambda_1 \Lambda_2 \Lambda_3} \int_0^{\varrho_3} \bar{\Lambda}_3(\lambda) G_\lambda d\lambda$$

$$= \left(1 - \frac{\tilde{F}_0}{F}\right)(\psi - \rho F) - \frac{\theta \kappa \gamma (\xi + \vartheta)(1 - \tilde{\mathfrak{R}}_0)}{\phi \xi \mu \Lambda_1 \Lambda_2 \Lambda_3} U$$

$$- \frac{\theta(\xi + \vartheta)}{\phi \xi \mu \Lambda_1 \Lambda_2 \Lambda_3} \left(\phi \sigma \left(1 - (\tilde{\mathfrak{R}}_{02} + \tilde{\mathfrak{R}}_{03})\right) + \kappa \varepsilon (1 - \tilde{\mathfrak{R}}_0)\right) UK.$$

After performing a direct calculation and utilizing the value $\tilde{F}_0 = \psi / \rho$, we acquire:

$$\frac{d\tilde{\Theta}_0}{dt} = \left(1 - \frac{\tilde{F}_0}{F}\right)(\rho \tilde{F}_0 - \rho F) - \frac{\theta \kappa \gamma (\xi + \vartheta)(1 - \tilde{\mathfrak{R}}_0)}{\phi \xi \mu \Lambda_1 \Lambda_2 \Lambda_3} U$$

$$- \frac{\theta(\xi + \vartheta)}{\phi \xi \mu \Lambda_1 \Lambda_2 \Lambda_3} \left(\phi \sigma \left(1 - (\tilde{\mathfrak{R}}_{02} + \tilde{\mathfrak{R}}_{03})\right) + \kappa \varepsilon (1 - \tilde{\mathfrak{R}}_0)\right) UK$$

$$= - \frac{\rho (F - \tilde{F}_0)^2}{F} - \frac{\theta \kappa \gamma (\xi + \vartheta)(1 - \tilde{\mathfrak{R}}_0)}{\phi \xi \mu \Lambda_1 \Lambda_2 \Lambda_3} U$$

$$- \frac{\theta(\xi + \vartheta)}{\phi \xi \mu \Lambda_1 \Lambda_2 \Lambda_3} \left(\phi \sigma \left(1 - (\tilde{\mathfrak{R}}_{02} + \tilde{\mathfrak{R}}_{03})\right) + \kappa \varepsilon (1 - \tilde{\mathfrak{R}}_0)\right) UK.$$

Clearly, $\frac{d\tilde{\Theta}_0}{dt} \leq 0$ when $\tilde{\mathfrak{R}}_0 < 1$. In addition, $\frac{d\tilde{\Theta}_0}{dt} = 0$ when $F = \tilde{F}_0$ and $K = U = 0$. Let $\bar{\Phi}_0 = \left\{ (F, Q, G, K, U) : \frac{d\tilde{\Theta}_0}{dt} = 0 \right\}$, and considering the largest invariant subset of $\bar{\Phi}_0$ as $\bar{\Phi}_0'$. Hence, all solutions converge to: $\bar{\Phi}_0'$. In $\bar{\Phi}_0'$, all elements satisfy $F(t) = \tilde{F}_0$ and $K(t) = U(t) = 0$. Subsequently, the fourth equation in system (23) yields

$$0 = \frac{dK(t)}{dt} = \mu \int_0^{\varrho_3} \bar{\Lambda}_3(\lambda) G_\lambda d\lambda.$$

The condition of nonnegativity for $G$ implies that $G(t)$ must be equal to zero for all values of $t$. Furthermore, the first equation of model (23) leads to

$$0 = \frac{dF(t)}{dt} = \psi - \rho \tilde{F}_0 - \varphi_2 \tilde{F}_0 Q(t) \implies Q(t) = 0, \text{ for all } t.$$

Then, $\bar{\Phi}_0' = \left\{ (F, Q, G, K, U) \in \bar{\Phi}_0 : F = \tilde{F}_0, Q = G = K = U = 0 \right\} = \{\Xi \tilde{Q}_0\}$. Therefore, based on L.I.P, we can conclude that $\Xi \tilde{Q}_0$ is G.A.S whenever $\tilde{\mathfrak{R}}_0 < 1$ [42].

In addition to this, model (23) can be rewritten as:

$$\frac{d\aleph(t)}{dt} = \mathcal{F}(\aleph(t), \aleph(t - \lambda)),$$

where $\aleph(t) = (F(t), Q(t), G(t), K(t), U(t))^T$. This system represents a coupled system of ordinary differential equations with a delay parameter. By employing total differentiation at the equilibrium point $\Xi \tilde{Q}_0$, we obtain:

$$\begin{cases} \frac{dF(t)}{dt} = \frac{\partial \mathcal{F}}{\partial F}|_{\Xi \tilde{Q}_0} F + \frac{\partial \mathcal{F}}{\partial Q}|_{\Xi \tilde{Q}_0} Q + \frac{\partial \mathcal{F}}{\partial G}|_{\Xi \tilde{Q}_0} G + \frac{\partial \mathcal{F}}{\partial K}|_{\Xi \tilde{Q}_0} K + \frac{\partial \mathcal{F}}{\partial U}|_{\Xi \tilde{Q}_0} U, \\ \frac{dQ(t)}{dt} = \frac{\partial \mathcal{F}}{\partial F}|_{\Xi \tilde{Q}_0} F + \frac{\partial \mathcal{F}}{\partial Q}|_{\Xi \tilde{Q}_0} Q + \frac{\partial \mathcal{F}}{\partial G}|_{\Xi \tilde{Q}_0} G + \frac{\partial \mathcal{F}}{\partial K}|_{\Xi \tilde{Q}_0} K + \frac{\partial \mathcal{F}}{\partial U}|_{\Xi \tilde{Q}_0} U, \\ \frac{dG(t)}{dt} = \frac{\partial \mathcal{F}}{\partial F}|_{\Xi \tilde{Q}_0} F + \frac{\partial \mathcal{F}}{\partial Q}|_{\Xi \tilde{Q}_0} Q + \frac{\partial \mathcal{F}}{\partial G}|_{\Xi \tilde{Q}_0} G + \frac{\partial \mathcal{F}}{\partial K}|_{\Xi \tilde{Q}_0} K + \frac{\partial \mathcal{F}}{\partial U}|_{\Xi \tilde{Q}_0} U, \\ \frac{dK(t)}{dt} = \frac{\partial \mathcal{F}}{\partial F}|_{\Xi \tilde{Q}_0} F + \frac{\partial \mathcal{F}}{\partial Q}|_{\Xi \tilde{Q}_0} Q + \frac{\partial \mathcal{F}}{\partial G}|_{\Xi \tilde{Q}_0} G + \frac{\partial \mathcal{F}}{\partial K}|_{\Xi \tilde{Q}_0} K + \frac{\partial \mathcal{F}}{\partial U}|_{\Xi \tilde{Q}_0} U, \\ \frac{dU(t)}{dt} = \frac{\partial \mathcal{F}}{\partial F}|_{\Xi \tilde{Q}_0} F + \frac{\partial \mathcal{F}}{\partial Q}|_{\Xi \tilde{Q}_0} Q + \frac{\partial \mathcal{F}}{\partial G}|_{\Xi \tilde{Q}_0} G + \frac{\partial \mathcal{F}}{\partial K}|_{\Xi \tilde{Q}_0} K + \frac{\partial \mathcal{F}}{\partial U}|_{\Xi \tilde{Q}_0} U. \end{cases} \quad (37)$$

Exponential solutions are a valid choice in linear DDEs because these equations exhibit linearity and bear resemblance to ordinary differential equations with constant coefficients. The preference for exponential solutions as an initial approach in linear systems stems from their ability to provide a clear and accessible means of characterizing system stability and dynamics. Assuming that the linear DDEs system (37) exhibits exponential solutions:

$$F = e^{xt} W_F, \quad Q = e^{xt} W_Q, \quad G = e^{xt} W_G, \quad K = e^{xt} W_K, \quad U = e^{xt} W_U.$$

Employing the above ansatz into system (37), and rearranging to obtain $AW = 0$, where

$$A = \begin{bmatrix} x+\rho & \varphi_2 \tilde{F}_0 & \varphi_3 \tilde{F}_0 & \varphi_1 \tilde{F}_0 & 0 \\ 0 & x+\xi+\vartheta-\varphi_2 \tilde{F}_0 \hat{\Lambda}_1 & -\varphi_3 \tilde{F}_0 \hat{\Lambda}_1 & -\varphi_1 \tilde{F}_0 \hat{\Lambda}_1 & 0 \\ 0 & -\xi \hat{\Lambda}_2 & x+\theta & 0 & 0 \\ 0 & 0 & -\mu \hat{\Lambda}_3 & x+\kappa & 0 \\ 0 & 0 & 0 & -\phi & x+\gamma \end{bmatrix}, \quad W = \begin{bmatrix} W_F \\ W_Q \\ W_G \\ W_K \\ W_U \end{bmatrix}.$$

It is essential to highlight that the characteristic equation arises when the matrix $A$ becomes non-invertible, a condition signified by the determinant $\det(A)$ equating to zero. Specifically, for the equilibrium point $\Xi \tilde{Q}_0$ in system (23), the characteristic equation takes the form $(x+\gamma)(x+\rho)\tilde{Y}(x) = 0$, with $\tilde{Y}(x)$ representing a continuous function defined on the interval $[0, \infty)$.

$$\begin{aligned} \tilde{Y}(x) = {} & x^3 + (\kappa+\xi+\vartheta+\theta-\tilde{F}_0 \hat{\Lambda}_1 \varphi_2) x^2 \\ & + (\theta\kappa-\xi \tilde{F}_0 \hat{\Lambda}_1 \hat{\Lambda}_2 \varphi_3 + (\theta+\kappa)(\xi+\vartheta-\tilde{F}_0 \hat{\Lambda}_1 \varphi_2)) x \\ & + \theta\kappa(\xi+\vartheta) - \tilde{F}_0 \hat{\Lambda}_1 (\xi\mu \hat{\Lambda}_2 \hat{\Lambda}_3 \varphi_1 + \kappa\theta\varphi_2 + \kappa\xi \hat{\Lambda}_2 \varphi_3), \end{aligned}$$

where $\hat{\Lambda}_i = \int_0^{\varrho_i} T_i(\lambda) e^{-(x+\alpha_i)\lambda} d\lambda$, $i = 1, 2, 3$. The case of $\tilde{\Re}_0 > 1$ implies that $\tilde{Y}(0) = \kappa\theta(\xi+\vartheta)(1-\tilde{\Re}_0) < 0$ and $\lim_{x\to\infty} \tilde{Y}(x) = \infty$, which guarantees that $\tilde{Y}(x)$ has a positive real root. Therefore, $\Xi \tilde{Q}_0$ is unstable when $\tilde{\Re}_0 > 1$. $\square$

**Theorem 5.** *For system (23), if $\tilde{\Re}_0 > 1$, then $\Xi \tilde{Q}_1$ is G.A.S.*

**Proof.** We consider the function $\tilde{\Theta}_1(F, Q, G, K, U)$ given by Equation (38) as:

$$\begin{aligned} \tilde{\Theta}_1 = {} & \tilde{F}_1 H\left(\frac{F}{\tilde{F}_1}\right) + \frac{\tilde{Q}_1}{\Lambda_1} H\left(\frac{Q}{\tilde{Q}_1}\right) + \frac{\tilde{F}_1(\Lambda_3 \varphi_1 \mu + \varphi_3(\kappa+\sigma \tilde{U}_1))\tilde{G}_1}{\theta(\kappa+\sigma \tilde{U}_1)} H\left(\frac{G}{\tilde{G}_1}\right) + \frac{\varphi_1 \tilde{F}_1 \tilde{K}_1}{\kappa+\sigma \tilde{U}_1} H\left(\frac{K}{\tilde{K}_1}\right) \\ & + \frac{\sigma\varphi_1 \tilde{F}_1}{2(\kappa+\sigma \tilde{U}_1)(\phi-\varepsilon \tilde{U}_1)} (U-\tilde{U}_1)^2 + \frac{\varphi_1 \tilde{F}_1 \tilde{K}_1}{\Lambda_1} \int_0^{\varrho_1} \bar{\Lambda}_1(\lambda) \int_{t-\lambda}^t H\left(\frac{F(\varkappa)K(\varkappa)}{\tilde{F}_1 \tilde{K}_1}\right) d\varkappa d\lambda \\ & + \frac{\varphi_2 \tilde{F}_1 \tilde{Q}_1}{\Lambda_1} \int_0^{\varrho_1} \bar{\Lambda}_1(\lambda) \int_{t-\lambda}^t H\left(\frac{F(\varkappa)Q(\varkappa)}{\tilde{F}_1 \tilde{Q}_1}\right) d\varkappa d\lambda + \frac{\varphi_3 \tilde{F}_1 \tilde{G}_1}{\Lambda_1} \int_0^{\varrho_1} \bar{\Lambda}_1(\lambda) \int_{t-\lambda}^t H\left(\frac{F(\varkappa)G(\varkappa)}{\tilde{F}_1 \tilde{G}_1}\right) d\varkappa d\lambda \\ & + \frac{\xi \tilde{F}_1 \tilde{Q}_1(\Lambda_3 \varphi_1 \mu + \varphi_3(\kappa+\sigma \tilde{U}_1))}{\theta(\kappa+\sigma \tilde{U}_1)} \int_0^{\varrho_2} \bar{\Lambda}_2(\lambda) \int_{t-\lambda}^t H\left(\frac{Q(\varkappa)}{\tilde{Q}_1}\right) d\varkappa d\lambda \\ & + \frac{\mu\varphi_1 \tilde{F}_1 \tilde{G}_1}{\kappa+\sigma \tilde{U}_1} \int_0^{\varrho_3} \bar{\Lambda}_3(\lambda) \int_{t-\lambda}^t H\left(\frac{G(\varkappa)}{\tilde{G}_1}\right) d\varkappa d\lambda. \end{aligned} \tag{38}$$

It is observed from the equilibrium condition Equation (30) that $\phi - \varepsilon \tilde{U}_1 = \frac{\gamma \tilde{U}_1}{\tilde{K}_1} > 0$.

It is clear that $\tilde{\Theta}_1$ is positive definite. Computing $\frac{d\tilde{\Theta}_1}{dt}$ along with the solutions of model (23), give us

$$\frac{d\tilde{\Theta}_1}{dt} = \left(1-\frac{\tilde{F}_1}{F}\right)\frac{dF}{dt} + \frac{1}{\Lambda_1}\left(1-\frac{\tilde{Q}_1}{Q}\right)\frac{dQ}{dt} + \frac{\tilde{F}_1(\Lambda_3 \varphi_1 \mu + \varphi_3(\kappa+\sigma \tilde{U}_1))}{\theta(\kappa+\sigma \tilde{U}_1)}\left(1-\frac{\tilde{G}_1}{G}\right)\frac{dG}{dt}$$

$$
+ \frac{\varphi_1 \tilde{F}_1}{\kappa + \sigma \tilde{U}_1} \left(1 - \frac{\tilde{K}_1}{K}\right) \frac{dK}{dt} + \frac{\sigma \varphi_1 \tilde{F}_1}{(\kappa + \sigma \tilde{U}_1)(\phi - \varepsilon \tilde{U}_1)} (U - \tilde{U}_1) \frac{dU}{dt}
$$

$$
+ \frac{\varphi_1 \tilde{F}_1 \tilde{K}_1}{\Lambda_1} \int_0^{\varrho_1} \bar{\Lambda}_1(\lambda) \left(\frac{FK}{\tilde{F}_1 \tilde{K}_1} - \frac{F_\lambda K_\lambda}{\tilde{F}_1 \tilde{K}_1} + \ln\left(\frac{F_\lambda K_\lambda}{FK}\right)\right) d\lambda
$$

$$
+ \frac{\varphi_2 \tilde{F}_1 \tilde{Q}_1}{\Lambda_1} \int_0^{\varrho_1} \bar{\Lambda}_1(\lambda) \left(\frac{FQ}{\tilde{F}_1 \tilde{Q}_1} - \frac{F_\lambda Q_\lambda}{\tilde{F}_1 \tilde{Q}_1} + \ln\left(\frac{F_\lambda Q_\lambda}{FQ}\right)\right) d\lambda
$$

$$
+ \frac{\varphi_3 \tilde{F}_1 \tilde{G}_1}{\Lambda_1} \int_0^{\varrho_1} \bar{\Lambda}_1(\lambda) \left(\frac{FG}{\tilde{F}_1 \tilde{G}_1} - \frac{F_\lambda G_\lambda}{\tilde{F}_1 \tilde{G}_1} + \ln\left(\frac{F_\lambda G_\lambda}{FG}\right)\right) d\lambda
$$

$$
+ \frac{\xi \tilde{F}_1 \tilde{Q}_1 (\Lambda_3 \varphi_1 \mu + \varphi_3(\kappa + \sigma \tilde{U}_1))}{\theta(\kappa + \sigma \tilde{U}_1)} \int_0^{\varrho_2} \bar{\Lambda}_2(\lambda) \left(\frac{Q}{\tilde{Q}_1} - \frac{Q_\lambda}{\tilde{Q}_1} + \ln\left(\frac{Q_\lambda}{Q}\right)\right) d\lambda
$$

$$
+ \frac{\mu \varphi_1 \tilde{F}_1 \tilde{G}_1}{\kappa + \sigma \tilde{U}_1} \int_0^{\varrho_3} \bar{\Lambda}_3(\lambda) \left(\frac{G}{\tilde{G}_1} - \frac{G_\lambda}{\tilde{G}_1} + \ln\left(\frac{G_\lambda}{G}\right)\right) d\lambda
$$

$$
= \left(1 - \frac{\tilde{F}_1}{F}\right)(\psi - \rho F - \varphi_1 FK - \varphi_2 FQ - \varphi_3 FG)
$$

$$
+ \frac{1}{\Lambda_1} \left(1 - \frac{\tilde{Q}_1}{Q}\right) \left(\int_0^{\varrho_1} \bar{\Lambda}_1(\lambda) F_\lambda (\varphi_1 K_\lambda + \varphi_2 Q_\lambda + \varphi_3 G_\lambda) d\lambda - (\xi + \vartheta) Q\right)
$$

$$
+ \frac{\tilde{F}_1 (\Lambda_3 \varphi_1 \mu + \varphi_3(\kappa + \sigma \tilde{U}_1))}{\theta(\kappa + \sigma \tilde{U}_1)} \left(1 - \frac{\tilde{G}_1}{G}\right) \left(\xi \int_0^{\varrho_2} \bar{\Lambda}_2(\lambda) Q_\lambda d\lambda - \theta G\right)
$$

$$
+ \frac{\varphi_1 \tilde{F}_1}{\kappa + \sigma \tilde{U}_1} \left(1 - \frac{\tilde{K}_1}{K}\right) \left(\mu \int_0^{\varrho_3} \bar{\Lambda}_3(\lambda) G_\lambda d\lambda - \kappa K - \sigma U K\right)
$$

$$
+ \frac{\sigma \varphi_1 \tilde{F}_1}{(\kappa + \sigma \tilde{U}_1)(\phi - \varepsilon \tilde{U}_1)} (U - \tilde{U}_1)(\phi K - \gamma U - \varepsilon U K)
$$

$$
+ \varphi_1 FK - \frac{\varphi_1 \tilde{F}_1 \tilde{K}_1}{\Lambda_1} \int_0^{\varrho_1} \bar{\Lambda}_1(\lambda) \left(\frac{F_\lambda K_\lambda}{\tilde{F}_1 \tilde{K}_1} - \ln\left(\frac{F_\lambda K_\lambda}{FK}\right)\right) d\lambda
$$

$$
+ \varphi_2 FQ - \frac{\varphi_2 \tilde{F}_1 \tilde{Q}_1}{\Lambda_1} \int_0^{\varrho_1} \bar{\Lambda}_1(\lambda) \left(\frac{F_\lambda Q_\lambda}{\tilde{F}_1 \tilde{Q}_1} - \ln\left(\frac{F_\lambda Q_\lambda}{FQ}\right)\right) d\lambda
$$

$$
+ \varphi_3 FG - \frac{\varphi_3 \tilde{F}_1 \tilde{G}_1}{\Lambda_1} \int_0^{\varrho_1} \bar{\Lambda}_1(\lambda) \left(\frac{F_\lambda G_\lambda}{\tilde{F}_1 \tilde{G}_1} - \ln\left(\frac{F_\lambda G_\lambda}{FG}\right)\right) d\lambda
$$

$$
+ \frac{\xi \tilde{F}_1 \tilde{Q}_1 (\Lambda_3 \varphi_1 \mu + \varphi_3(\kappa + \sigma \tilde{U}_1))}{\theta(\kappa + \sigma \tilde{U}_1)} \left(\frac{\Lambda_2 Q}{\tilde{Q}_1} - \int_0^{\varrho_2} \bar{\Lambda}_2(\lambda) \left(\frac{Q_\lambda}{\tilde{Q}_1} - \ln\left(\frac{Q_\lambda}{Q}\right)\right) d\lambda\right)
$$

$$
+ \frac{\mu \varphi_1 \tilde{F}_1 \tilde{G}_1}{\kappa + \sigma \tilde{U}_1} \left(\frac{\Lambda_3 G}{\tilde{G}_1} - \int_0^{\varrho_3} \bar{\Lambda}_3(\lambda) \left(\frac{G_\lambda}{\tilde{G}_1} - \ln\left(\frac{G_\lambda}{G}\right)\right) d\lambda\right).
$$

This implies that

$$
\frac{d\tilde{\Theta}_1}{dt} = \left(1 - \frac{\tilde{F}_1}{F}\right)(\psi - \rho F) + \left(\varphi_2 \tilde{F}_1 - \frac{\xi + \vartheta}{\Lambda_1} + \frac{\xi \Lambda_2 \tilde{F}_1 (\Lambda_3 \varphi_1 \mu + \varphi_3(\kappa + \sigma \tilde{U}_1))}{\theta(\kappa + \sigma \tilde{U}_1)}\right) Q
$$

$$
+ \varphi_1 \tilde{F}_1 K + \left(\varphi_3 \tilde{F}_1 - \frac{\tilde{F}_1 (\Lambda_3 \varphi_1 \mu + \varphi_3(\kappa + \sigma \tilde{U}_1))}{\kappa + \sigma \tilde{U}_1} + \frac{\mu \varphi_1 \Lambda_3 \tilde{F}_1}{\kappa + \sigma \tilde{U}_1}\right) G
$$

$$
- \frac{1}{\Lambda_1} \int_0^{\varrho_1} \bar{\Lambda}_1(\lambda) \frac{F_\lambda \tilde{Q}_1}{Q} (\varphi_1 K_\lambda + \varphi_2 Q_\lambda + \varphi_3 G_\lambda) d\lambda + \frac{(\xi + \vartheta) \tilde{Q}_1}{\Lambda_1}
$$

$$
- \frac{\xi \tilde{F}_1 (\Lambda_3 \varphi_1 \mu + \varphi_3(\kappa + \sigma \tilde{U}_1))}{\theta(\kappa + \sigma \tilde{U}_1)} \int_0^{\varrho_2} \bar{\Lambda}_2(\lambda) \frac{Q_\lambda \tilde{G}_1}{G} d\lambda + \frac{\tilde{F}_1 (\Lambda_3 \varphi_1 \mu + \varphi_3(\kappa + \sigma \tilde{U}_1))}{\kappa + \sigma \tilde{U}_1} \tilde{G}_1
$$

$$
- \frac{\mu \varphi_1 \tilde{F}_1}{\kappa + \sigma \tilde{U}_1} \int_0^{\varrho_3} \bar{\Lambda}_3(\lambda) \frac{G_\lambda \tilde{K}_1}{K} d\lambda - \frac{\kappa \varphi_1 \tilde{F}_1}{\kappa + \sigma \tilde{U}_1} (K - \tilde{K}_1) - \frac{\sigma \varphi_1 \tilde{F}_1}{\kappa + \sigma \tilde{U}_1} U(K - \tilde{K}_1)
$$

$$
+ \frac{\sigma \varphi_1 \tilde{F}_1}{(\kappa + \sigma \tilde{U}_1)(\phi - \varepsilon \tilde{U}_1)} (U - \tilde{U}_1)(\phi K - \gamma U - \varepsilon U K) + \frac{\varphi_1 \tilde{F}_1 \tilde{K}_1}{\Lambda_1} \int_0^{\varrho_1} \bar{\Lambda}_1(\lambda) \ln\left(\frac{F_\lambda K_\lambda}{FK}\right) d\lambda
$$

$$+ \frac{\varphi_2 \tilde{F}_1 \tilde{Q}_1}{\Lambda_1} \int_0^{\varrho_1} \bar{\Lambda}_1(\lambda) \ln\left(\frac{F_\lambda Q_\lambda}{FQ}\right) d\lambda + \frac{\varphi_3 \tilde{F}_1 \tilde{G}_1}{\Lambda_1} \int_0^{\varrho_1} \bar{\Lambda}_1(\lambda) \ln\left(\frac{F_\lambda G_\lambda}{FG}\right) d\lambda$$

$$+ \frac{\xi \tilde{F}_1 \tilde{Q}_1 (\Lambda_3 \varphi_1 \mu + \varphi_3(\kappa + \sigma \tilde{U}_1))}{\theta(\kappa + \sigma \tilde{U}_1)} \int_0^{\varrho_2} \bar{\Lambda}_2(\lambda) \ln\left(\frac{Q_\lambda}{Q}\right) d\lambda + \frac{\mu \varphi_1 \tilde{F}_1 \tilde{G}_1}{\kappa + \sigma \tilde{U}_1} \int_0^{\varrho_3} \bar{\Lambda}_3(\lambda) \ln\left(\frac{G_\lambda}{G}\right) d\lambda. \qquad (39)$$

Using the following equilibrium conditions for $\Xi \tilde{Q}_1$

$$\psi = \rho \tilde{F}_1 + \varphi_1 \tilde{F}_1 \tilde{K}_1 + \varphi_2 \tilde{F}_1 \tilde{Q}_1 + \varphi_3 \tilde{F}_1 \tilde{G}_1,$$

$$\varphi_1 \tilde{F}_1 \tilde{K}_1 + \varphi_2 \tilde{F}_1 \tilde{Q}_1 + \varphi_3 \tilde{F}_1 \tilde{G}_1 = \frac{(\xi + \vartheta) \tilde{Q}_1}{\Lambda_1},$$

$$\tilde{Q}_1 = \frac{\theta \tilde{G}_1}{\xi \Lambda_2} \Longrightarrow \tilde{G}_1 = \frac{\xi \Lambda_2 \tilde{Q}_1}{\theta},$$

$$\tilde{K}_1 = \frac{\mu \Lambda_3 \tilde{G}_1}{\kappa + \sigma \tilde{U}_1} = \frac{\xi \mu \Lambda_2 \Lambda_3 \tilde{Q}_1}{\theta(\kappa + \sigma \tilde{U}_1)},$$

$$\phi \tilde{K}_1 = \gamma \tilde{U}_1 + \varepsilon \tilde{U}_1 \tilde{K}_1,$$

we get

$$\varphi_1 \tilde{F}_1 \tilde{K}_1 + \varphi_3 \tilde{F}_1 \tilde{G}_1 = \frac{\tilde{F}_1 \tilde{K}_1 (\mu \Lambda_3 \varphi_1 + \varphi_3(\kappa + \sigma \tilde{U}_1))}{\Lambda_3 \mu} = \frac{\xi \Lambda_2 \tilde{F}_1 \tilde{Q}_1 (\mu \Lambda_3 \varphi_1 + \varphi_3(\kappa + \sigma \tilde{U}_1))}{\theta(\kappa + \sigma \tilde{U}_1)},$$

$$\left( \varphi_2 \tilde{F}_1 - \frac{\xi + \vartheta}{\Lambda_1} + \frac{\xi \Lambda_2 \tilde{F}_1 (\Lambda_3 \varphi_1 \mu + \varphi_3(\kappa + \sigma \tilde{U}_1))}{\theta(\kappa + \sigma \tilde{U}_1)} \right) \tilde{Q}_1 = 0.$$

Therefore, Equation (39) will be represented in the following manner:

$$\frac{d \tilde{\Theta}_1}{dt} = \left(1 - \frac{\tilde{F}_1}{F}\right) (\rho \tilde{F}_1 - \rho F) + \left(\varphi_1 \tilde{F}_1 \tilde{K}_1 + \varphi_2 \tilde{F}_1 \tilde{Q}_1 + \varphi_3 \tilde{F}_1 \tilde{G}_1\right) \left(1 - \frac{\tilde{F}_1}{F}\right) + \varphi_1 \tilde{F}_1 K$$

$$- \frac{1}{\Lambda_1} \int_0^{\varrho_1} \bar{\Lambda}_1(\lambda) \frac{F_\lambda \tilde{Q}_1}{Q} (\varphi_1 K_\lambda + \varphi_2 Q_\lambda + \varphi_3 G_\lambda) d\lambda + \varphi_1 \tilde{F}_1 \tilde{K}_1 + \varphi_2 \tilde{F}_1 \tilde{Q}_1 + \varphi_3 \tilde{F}_1 \tilde{G}_1$$

$$- \frac{\xi \tilde{F}_1 \tilde{Q}_1 (\Lambda_3 \varphi_1 \mu + \varphi_3(\kappa + \sigma \tilde{U}_1))}{\theta(\kappa + \sigma \tilde{U}_1)} \int_0^{\varrho_2} \bar{\Lambda}_2(\lambda) \frac{Q_\lambda \tilde{G}_1}{\tilde{Q}_1 G} d\lambda + \varphi_1 \tilde{F}_1 \tilde{K}_1 + \varphi_3 \tilde{F}_1 \tilde{G}_1$$

$$- \frac{\mu \varphi_1 \tilde{F}_1 \tilde{G}_1}{\kappa + \sigma \tilde{U}_1} \int_0^{\varrho_3} \bar{\Lambda}_3(\lambda) \frac{G_\lambda \tilde{K}_1}{\tilde{G}_1 K} d\lambda - \frac{\kappa \varphi_1 \tilde{F}_1}{\kappa + \sigma \tilde{U}_1} (K - \tilde{K}_1)$$

$$- \frac{\sigma \varphi_1 \tilde{F}_1}{\kappa + \sigma \tilde{U}_1} U(K - \tilde{K}_1) + \frac{\sigma \varphi_1 \tilde{F}_1}{\kappa + \sigma \tilde{U}_1} \tilde{U}_1(K - \tilde{K}_1) - \frac{\sigma \varphi_1 \tilde{F}_1}{\kappa + \sigma \tilde{U}_1} \tilde{U}_1(K - \tilde{K}_1)$$

$$+ \frac{\sigma \varphi_1 \tilde{F}_1}{(\kappa + \sigma \tilde{U}_1)(\phi - \varepsilon \tilde{U}_1)} (U - \tilde{U}_1)(\phi K - \gamma U - \varepsilon UK - \phi \tilde{K}_1 + \gamma \tilde{U}_1 + \varepsilon \tilde{U}_1 \tilde{K}_1 - \varepsilon \tilde{U}_1 K + \varepsilon \tilde{U}_1 K)$$

$$+ \frac{\varphi_1 \tilde{F}_1 \tilde{K}_1}{\Lambda_1} \int_0^{\varrho_1} \bar{\Lambda}_1(\lambda) \ln\left(\frac{F_\lambda K_\lambda}{FK}\right) d\lambda + \frac{\varphi_2 \tilde{F}_1 \tilde{Q}_1}{\Lambda_1} \int_0^{\varrho_1} \bar{\Lambda}_1(\lambda) \ln\left(\frac{F_\lambda Q_\lambda}{FQ}\right) d\lambda$$

$$+ \frac{\varphi_3 \tilde{F}_1 \tilde{G}_1}{\Lambda_1} \int_0^{\varrho_1} \bar{\Lambda}_1(\lambda) \ln\left(\frac{F_\lambda G_\lambda}{FG}\right) d\lambda$$

$$+ \frac{\xi \tilde{F}_1 \tilde{Q}_1 (\Lambda_3 \varphi_1 \mu + \varphi_3(\kappa + \sigma \tilde{U}_1))}{\theta(\kappa + \sigma \tilde{U}_1)} \int_0^{\varrho_2} \bar{\Lambda}_2(\lambda) \ln\left(\frac{Q_\lambda}{Q}\right) d\lambda$$

$$+ \frac{\mu \varphi_1 \tilde{F}_1 \tilde{G}_1}{(\kappa + \sigma \tilde{U}_1)} \int_0^{\varrho_3} \bar{\Lambda}_3(\lambda) \ln\left(\frac{G_\lambda}{G}\right) d\lambda$$

$$= - \frac{\rho(F - \tilde{F}_1)^2}{F} + \left(\varphi_1 \tilde{F}_1 \tilde{K}_1 + \varphi_2 \tilde{F}_1 \tilde{Q}_1 + \varphi_3 \tilde{F}_1 \tilde{G}_1\right) \left(2 - \frac{\tilde{F}_1}{F}\right) + \varphi_1 \tilde{F}_1 K$$

$$- \frac{\varphi_1 \tilde{F}_1 \tilde{K}_1}{\Lambda_1} \int_0^{\varrho_1} \bar{\Lambda}_1(\lambda) \frac{F_\lambda K_\lambda \tilde{Q}_1}{\tilde{F}_1 \tilde{K}_1 Q} d\lambda - \frac{\varphi_2 \tilde{F}_1 \tilde{Q}_1}{\Lambda_1} \int_0^{\varrho_1} \bar{\Lambda}_1(\lambda) \frac{F_\lambda Q_\lambda}{\tilde{F}_1 Q} d\lambda$$

$$
-\frac{\varphi_3\tilde{F}_1\tilde{G}_1}{\Lambda_1}\int_0^{\varrho_1}\bar{\Lambda}_1(\lambda)\frac{F_\lambda G_\lambda \tilde{Q}_1}{\tilde{F}_1\tilde{G}_1 Q}d\lambda - \frac{\varphi_1\tilde{F}_1\tilde{K}_1 + \varphi_3\tilde{F}_1\tilde{G}_1}{\Lambda_2}\int_0^{\varrho_2}\bar{\Lambda}_2(\lambda)\frac{Q_\lambda \tilde{G}_1}{\tilde{Q}_1 G}d\lambda
$$

$$
+ \varphi_1\tilde{F}_1\tilde{K}_1 + \varphi_3\tilde{F}_1\tilde{G}_1 - \frac{\varphi_1\tilde{F}_1\tilde{K}_1}{\Lambda_3}\int_0^{\varrho_3}\bar{\Lambda}_3(\lambda)\frac{G_\lambda \tilde{K}_1}{\tilde{G}_1 K}d\lambda
$$

$$
-\frac{\varphi_1\tilde{F}_1(\kappa + \sigma\tilde{U}_1)}{\kappa + \sigma\tilde{U}_1}(K - \tilde{K}_1) - \frac{\sigma\varphi_1\tilde{F}_1}{\kappa + \sigma\tilde{U}_1}(U - \tilde{U}_1)(K - \tilde{K}_1)
$$

$$
+ \frac{\sigma\varphi_1\tilde{F}_1(\phi - \varepsilon\tilde{U}_1)}{(\kappa + \sigma\tilde{U}_1)(\phi - \varepsilon\tilde{U}_1)}(U - \tilde{U}_1)(K - \tilde{K}_1) - \frac{\sigma\varphi_1\tilde{F}_1(\gamma + \varepsilon K)}{(\kappa + \sigma\tilde{U}_1)(\phi - \varepsilon\tilde{U}_1)}(U - \tilde{U}_1)^2
$$

$$
+ \frac{\varphi_1\tilde{F}_1\tilde{K}_1}{\Lambda_1}\int_0^{\varrho_1}\bar{\Lambda}_1(\lambda)\ln\left(\frac{F_\lambda K_\lambda}{FK}\right)d\lambda + \frac{\varphi_2\tilde{F}_1\tilde{Q}_1}{\Lambda_1}\int_0^{\varrho_1}\bar{\Lambda}_1(\lambda)\ln\left(\frac{F_\lambda Q_\lambda}{FQ}\right)d\lambda
$$

$$
+ \frac{\varphi_3\tilde{F}_1\tilde{G}_1}{\Lambda_1}\int_0^{\varrho_1}\bar{\Lambda}_1(\lambda)\ln\left(\frac{F_\lambda G_\lambda}{FG}\right)d\lambda + \frac{\varphi_1\tilde{F}_1\tilde{K}_1 + \varphi_3\tilde{F}_1\tilde{G}_1}{\Lambda_2}\int_0^{\varrho_2}\bar{\Lambda}_2(\lambda)\ln\left(\frac{Q_\lambda}{Q}\right)d\lambda
$$

$$
+ \frac{\varphi_1\tilde{F}_1\tilde{K}_1}{\Lambda_3}\int_0^{\varrho_3}\bar{\Lambda}_3(\lambda)\ln\left(\frac{G_\lambda}{G}\right)d\lambda
$$

$$
= -\frac{\rho(F - \tilde{F}_1)^2}{F} + \left(\varphi_1\tilde{F}_1\tilde{K}_1 + \varphi_2\tilde{F}_1\tilde{Q}_1 + \varphi_3\tilde{F}_1\tilde{G}_1\right)\left(2 - \frac{\tilde{F}_1}{F}\right) + \varphi_1\tilde{F}_1 K
$$

$$
- \frac{\varphi_1\tilde{F}_1\tilde{K}_1}{\Lambda_1}\int_0^{\varrho_1}\bar{\Lambda}_1(\lambda)\frac{F_\lambda K_\lambda \tilde{Q}_1}{\tilde{F}_1\tilde{K}_1 Q}d\lambda - \frac{\varphi_2\tilde{F}_1\tilde{Q}_1}{\Lambda_1}\int_0^{\varrho_1}\bar{\Lambda}_1(\lambda)\frac{F_\lambda Q_\lambda}{\tilde{F}_1 Q}d\lambda
$$

$$
- \frac{\varphi_3\tilde{F}_1\tilde{G}_1}{\Lambda_1}\int_0^{\varrho_1}\bar{\Lambda}_1(\lambda)\frac{F_\lambda G_\lambda \tilde{Q}_1}{\tilde{F}_1\tilde{G}_1 Q}d\lambda - \frac{\varphi_1\tilde{F}_1\tilde{K}_1 + \varphi_3\tilde{F}_1\tilde{G}_1}{\Lambda_2}\int_0^{\varrho_2}\bar{\Lambda}_2(\lambda)\frac{Q_\lambda \tilde{G}_1}{\tilde{Q}_1 G}d\lambda
$$

$$
+ \varphi_1\tilde{F}_1\tilde{K}_1 + \varphi_3\tilde{F}_1\tilde{G}_1 - \frac{\varphi_1\tilde{F}_1\tilde{K}_1}{\Lambda_3}\int_0^{\varrho_3}\bar{\Lambda}_3(\lambda)\frac{G_\lambda \tilde{K}_1}{\tilde{G}_1 K}d\lambda - \varphi_1\tilde{F}_1\tilde{K}_1\left(\frac{K}{\tilde{K}_1} - 1\right)
$$

$$
- \frac{\sigma\varphi_1\tilde{F}_1(\gamma + \varepsilon K)}{(\kappa + \sigma\tilde{U}_1)(\phi - \varepsilon\tilde{U}_1)}(U - \tilde{U}_1)^2 + \frac{\varphi_1\tilde{F}_1\tilde{K}_1}{\Lambda_1}\int_0^{\varrho_1}\bar{\Lambda}_1(\lambda)\ln\left(\frac{F_\lambda K_\lambda}{FK}\right)d\lambda
$$

$$
+ \frac{\varphi_2\tilde{F}_1\tilde{Q}_1}{\Lambda_1}\int_0^{\varrho_1}\bar{\Lambda}_1(\lambda)\ln\left(\frac{F_\lambda Q_\lambda}{FQ}\right)d\lambda + \frac{\varphi_3\tilde{F}_1\tilde{G}_1}{\Lambda_1}\int_0^{\varrho_1}\bar{\Lambda}_1(\lambda)\ln\left(\frac{F_\lambda G_\lambda}{FG}\right)d\lambda
$$

$$
+ \frac{\varphi_1\tilde{F}_1\tilde{K}_1 + \varphi_3\tilde{F}_1\tilde{G}_1}{\Lambda_2}\int_0^{\varrho_2}\bar{\Lambda}_2(\lambda)\ln\left(\frac{Q_\lambda}{Q}\right)d\lambda + \frac{\varphi_1\tilde{F}_1\tilde{K}_1}{\Lambda_3}\int_0^{\varrho_3}\bar{\Lambda}_3(\lambda)\ln\left(\frac{G_\lambda}{G}\right)d\lambda.
$$

Therefore

$$
\frac{d\tilde{\Theta}_1}{dt} = -\frac{\rho(F - \tilde{F}_1)^2}{F} + \left(\varphi_1\tilde{F}_1\tilde{K}_1 + \varphi_2\tilde{F}_1\tilde{Q}_1 + \varphi_3\tilde{F}_1\tilde{G}_1\right)\left(2 - \frac{\tilde{F}_1}{F}\right)
$$

$$
- \frac{\varphi_1\tilde{F}_1\tilde{K}_1}{\Lambda_1}\int_0^{\varrho_1}\bar{\Lambda}_1(\lambda)\frac{F_\lambda K_\lambda \tilde{Q}_1}{\tilde{F}_1\tilde{K}_1 Q}d\lambda - \frac{\varphi_2\tilde{F}_1\tilde{Q}_1}{\Lambda_1}\int_0^{\varrho_1}\bar{\Lambda}_1(\lambda)\frac{F_\lambda Q_\lambda}{\tilde{F}_1 Q}d\lambda
$$

$$
- \frac{\varphi_3\tilde{F}_1\tilde{G}_1}{\Lambda_1}\int_0^{\varrho_1}\bar{\Lambda}_1(\lambda)\frac{F_\lambda G_\lambda \tilde{Q}_1}{\tilde{F}_1\tilde{G}_1 Q}d\lambda - \frac{\varphi_1\tilde{F}_1\tilde{K}_1 + \varphi_3\tilde{F}_1\tilde{G}_1}{\Lambda_2}\int_0^{\varrho_2}\bar{\Lambda}_2(\lambda)\frac{Q_\lambda \tilde{G}_1}{\tilde{Q}_1 G}d\lambda
$$

$$
+ \varphi_1\tilde{F}_1\tilde{K}_1 + \varphi_3\tilde{F}_1\tilde{G}_1 - \frac{\varphi_1\tilde{F}_1\tilde{K}_1}{\Lambda_3}\int_0^{\varrho_3}\bar{\Lambda}_3(\lambda)\frac{G_\lambda \tilde{K}_1}{\tilde{G}_1 K}d\lambda + \varphi_1\tilde{F}_1\tilde{K}_1
$$

$$
- \frac{\sigma\varphi_1\tilde{F}_1(\gamma + \varepsilon K)}{(\kappa + \sigma\tilde{U}_1)(\phi - \varepsilon\tilde{U}_1)}(U - \tilde{U}_1)^2 + \frac{\varphi_1\tilde{F}_1\tilde{K}_1}{\Lambda_1}\int_0^{\varrho_1}\bar{\Lambda}_1(\lambda)\ln\left(\frac{F_\lambda K_\lambda}{FK}\right)d\lambda
$$

$$
+ \frac{\varphi_2\tilde{F}_1\tilde{Q}_1}{\Lambda_1}\int_0^{\varrho_1}\bar{\Lambda}_1(\lambda)\ln\left(\frac{F_\lambda Q_\lambda}{FQ}\right)d\lambda + \frac{\varphi_3\tilde{F}_1\tilde{G}_1}{\Lambda_1}\int_0^{\varrho_1}\bar{\Lambda}_1(\lambda)\ln\left(\frac{F_\lambda G_\lambda}{FG}\right)d\lambda
$$

$$
+ \frac{\varphi_1\tilde{F}_1\tilde{K}_1 + \varphi_3\tilde{F}_1\tilde{G}_1}{\Lambda_2}\int_0^{\varrho_2}\bar{\Lambda}_2(\lambda)\ln\left(\frac{Q_\lambda}{Q}\right)d\lambda + \frac{\varphi_1\tilde{F}_1\tilde{K}_1}{\Lambda_3}\int_0^{\varrho_3}\bar{\Lambda}_3(\lambda)\ln\left(\frac{G_\lambda}{G}\right)d\lambda.
$$

Furthermore, we have

$$\ln\left(\frac{F_\lambda K_\lambda}{FK}\right) = \ln\left(\frac{F_\lambda K_\lambda \tilde{Q}_1}{\tilde{F}_1 \tilde{K}_1 Q}\right) + \ln\left(\frac{\tilde{F}_1}{F}\right) + \ln\left(\frac{\tilde{K}_1 Q}{K \tilde{Q}_1}\right),$$

$$\ln\left(\frac{F_\lambda G_\lambda}{FG}\right) = \ln\left(\frac{F_\lambda G_\lambda \tilde{Q}_1}{\tilde{F}_1 \tilde{G}_1 Q}\right) + \ln\left(\frac{\tilde{F}_1}{F}\right) + \ln\left(\frac{\tilde{G}_1 Q}{G \tilde{Q}_1}\right),$$

$$\ln\left(\frac{F_\lambda Q_\lambda}{FQ}\right) = \ln\left(\frac{F_\lambda Q_\lambda}{\tilde{F}_1 Q}\right) + \ln\left(\frac{\tilde{F}_1}{F}\right),$$

$$\ln\left(\frac{Q_\lambda}{Q}\right) = \ln\left(\frac{Q_\lambda \tilde{G}_1}{\tilde{Q}_1 G}\right) + \ln\left(\frac{\tilde{Q}_1 G}{Q \tilde{G}_1}\right),$$

$$\ln\left(\frac{G_\lambda}{G}\right) = \ln\left(\frac{G_\lambda \tilde{K}_1}{\tilde{G}_1 K}\right) + \ln\left(\frac{\tilde{G}_1 K}{G \tilde{K}_1}\right).$$

Then, $\frac{d\tilde{\Theta}_1}{dt}$ will be

$$\begin{aligned}
\frac{d\tilde{\Theta}_1}{dt} =\ & -\frac{\rho(F - \tilde{F}_1)^2}{F} + \left(\varphi_1 \tilde{F}_1 \tilde{K}_1 + \varphi_2 \tilde{F}_1 \tilde{Q}_1 + \varphi_3 \tilde{F}_1 \tilde{G}_1\right)\left(2 - \frac{\tilde{F}_1}{F}\right) + \varphi_1 \tilde{F}_1 \tilde{K}_1 \\
& - \frac{\sigma \varphi_1 \tilde{F}_1 (\gamma + \varepsilon K)}{(\kappa + \sigma \tilde{U}_1)(\phi - \varepsilon \tilde{U}_1)}(U - \tilde{U}_1)^2 - \frac{\varphi_1 \tilde{F}_1 \tilde{K}_1}{\Lambda_1} \int_0^{\varrho_1} \bar{\Lambda}_1(\lambda) \frac{F_\lambda K_\lambda \tilde{Q}_1}{\tilde{F}_1 \tilde{K}_1 Q} d\lambda \\
& - \frac{\varphi_2 \tilde{F}_1 \tilde{Q}_1}{\Lambda_1} \int_0^{\varrho_1} \bar{\Lambda}_1(\lambda) \frac{F_\lambda Q_\lambda}{\tilde{F}_1 Q} d\lambda - \frac{\varphi_3 \tilde{F}_1 \tilde{G}_1}{\Lambda_1} \int_0^{\varrho_1} \bar{\Lambda}_1(\lambda) \frac{F_\lambda G_\lambda \tilde{Q}_1}{\tilde{F}_1 \tilde{G}_1 Q} d\lambda \\
& - \frac{\varphi_1 \tilde{F}_1 \tilde{K}_1 + \varphi_3 \tilde{F}_1 \tilde{G}_1}{\Lambda_2} \int_0^{\varrho_2} \bar{\Lambda}_2(\lambda) \frac{Q_\lambda \tilde{G}_1}{\tilde{Q}_1 G} d\lambda + \varphi_1 \tilde{F}_1 \tilde{K}_1 + \varphi_3 \tilde{F}_1 \tilde{G}_1 \\
& - \frac{\varphi_1 \tilde{F}_1 \tilde{K}_1}{\Lambda_3} \int_0^{\varrho_3} \bar{\Lambda}_3(\lambda) \frac{G_\lambda \tilde{K}_1}{\tilde{G}_1 K} d\lambda \\
& + \frac{\varphi_1 \tilde{F}_1 \tilde{K}_1}{\Lambda_1} \int_0^{\varrho_1} \bar{\Lambda}_1(\lambda) \left[\ln\left(\frac{F_\lambda K_\lambda \tilde{Q}_1}{\tilde{F}_1 \tilde{K}_1 Q}\right) + \ln\left(\frac{\tilde{F}_1}{F}\right) + \ln\left(\frac{\tilde{K}_1 Q}{K \tilde{Q}_1}\right)\right] d\lambda \\
& + \frac{\varphi_2 \tilde{F}_1 \tilde{Q}_1}{\Lambda_1} \int_0^{\varrho_1} \bar{\Lambda}_1(\lambda) \left[\ln\left(\frac{F_\lambda Q_\lambda}{\tilde{F}_1 Q}\right) + \ln\left(\frac{\tilde{F}_1}{F}\right)\right] d\lambda \\
& + \frac{\varphi_3 \tilde{F}_1 \tilde{G}_1}{\Lambda_1} \int_0^{\varrho_1} \bar{\Lambda}_1(\lambda) \left[\ln\left(\frac{F_\lambda G_\lambda \tilde{Q}_1}{\tilde{F}_1 \tilde{G}_1 Q}\right) + \ln\left(\frac{\tilde{F}_1}{F}\right) + \ln\left(\frac{\tilde{G}_1 Q}{G \tilde{Q}_1}\right)\right] d\lambda \\
& + \frac{\varphi_1 \tilde{F}_1 \tilde{K}_1 + \varphi_3 \tilde{F}_1 \tilde{G}_1}{\Lambda_2} \int_0^{\varrho_2} \bar{\Lambda}_2(\lambda) \left[\ln\left(\frac{Q_\lambda \tilde{G}_1}{\tilde{Q}_1 G}\right) + \ln\left(\frac{\tilde{Q}_1 G}{Q \tilde{G}_1}\right)\right] d\lambda \\
& + \frac{\varphi_1 \tilde{F}_1 \tilde{K}_1}{\Lambda_3} \int_0^{\varrho_3} \bar{\Lambda}_3(\lambda) \left[\ln\left(\frac{G_\lambda \tilde{K}_1}{\tilde{G}_1 K}\right) + \ln\left(\frac{\tilde{G}_1 K}{G \tilde{K}_1}\right)\right] d\lambda \\
=\ & -\frac{\rho(F - \tilde{F}_1)^2}{F} - \left(\varphi_1 \tilde{F}_1 \tilde{K}_1 + \varphi_2 \tilde{F}_1 \tilde{Q}_1 + \varphi_3 \tilde{F}_1 \tilde{G}_1\right)\left[\frac{\tilde{F}_1}{F} - 1 - \ln\left(\frac{\tilde{F}_1}{F}\right)\right] \\
& - \frac{\varphi_1 \tilde{F}_1 \tilde{K}_1}{\Lambda_1} \int_0^{\varrho_1} \bar{\Lambda}_1(\lambda) \left[\frac{F_\lambda K_\lambda \tilde{Q}_1}{\tilde{F}_1 \tilde{K}_1 Q} - 1 - \ln\left(\frac{F_\lambda K_\lambda \tilde{Q}_1}{\tilde{F}_1 \tilde{K}_1 Q}\right)\right] d\lambda \\
& - \frac{\varphi_2 \tilde{F}_1 \tilde{Q}_1}{\Lambda_1} \int_0^{\varrho_1} \bar{\Lambda}_1(\lambda) \left[\frac{F_\lambda Q_\lambda}{\tilde{F}_1 Q} - 1 - \ln\left(\frac{F_\lambda Q_\lambda}{\tilde{F}_1 Q}\right)\right] d\lambda \\
& - \frac{\varphi_3 \tilde{F}_1 \tilde{G}_1}{\Lambda_1} \int_0^{\varrho_1} \bar{\Lambda}_1(\lambda) \left[\frac{F_\lambda G_\lambda \tilde{Q}_1}{\tilde{F}_1 \tilde{G}_1 Q} - 1 - \ln\left(\frac{F_\lambda G_\lambda \tilde{Q}_1}{\tilde{F}_1 \tilde{G}_1 Q}\right)\right] d\lambda \\
& - \frac{\varphi_1 \tilde{F}_1 \tilde{K}_1 + \varphi_3 \tilde{F}_1 \tilde{G}_1}{\Lambda_2} \int_0^{\varrho_2} \bar{\Lambda}_2(\lambda) \left[\frac{Q_\lambda \tilde{G}_1}{\tilde{Q}_1 G} - 1 - \ln\left(\frac{Q_\lambda \tilde{G}_1}{\tilde{Q}_1 G}\right)\right] d\lambda \\
& - \frac{\varphi_1 \tilde{F}_1 \tilde{K}_1}{\Lambda_3} \int_0^{\varrho_3} \bar{\Lambda}_3(\lambda) \left[\frac{G_\lambda \tilde{K}_1}{\tilde{G}_1 K} - 1 - \ln\left(\frac{G_\lambda \tilde{K}_1}{\tilde{G}_1 K}\right)\right] d\lambda
\end{aligned}$$

$$- \frac{\sigma \varphi_1 \tilde{F}_1 (\gamma + \varepsilon K)}{(\kappa + \sigma \tilde{U}_1)(\phi - \varepsilon \tilde{U}_1)} (U - \tilde{U}_1)^2.$$

Simplifying the result, we obtain

$$
\begin{aligned}
\frac{d\tilde{\Theta}_1}{dt} = & -\frac{\rho (F - \tilde{F}_1)^2}{F} - \frac{\varphi_1 \tilde{F}_1 \tilde{K}_1}{\Lambda_1} \int_0^{\varrho_1} \bar{\Lambda}_1(\lambda) \left[ H\left( \frac{F_\lambda K_\lambda \tilde{Q}_1}{\tilde{F}_1 \tilde{K}_1 Q} \right) + H\left( \frac{\tilde{F}_1}{F} \right) \right] d\lambda \\
& - \frac{\varphi_2 \tilde{F}_1 \tilde{Q}_1}{\Lambda_1} \int_0^{\varrho_1} \bar{\Lambda}_1(\lambda) \left[ H\left( \frac{F_\lambda Q_\lambda}{\tilde{F}_1 Q} \right) + H\left( \frac{\tilde{F}_1}{F} \right) \right] d\lambda \\
& - \frac{\varphi_3 \tilde{F}_1 \tilde{G}_1}{\Lambda_1} \int_0^{\varrho_1} \bar{\Lambda}_1(\lambda) \left[ H\left( \frac{F_\lambda G_\lambda \tilde{Q}_1}{\tilde{F}_1 \tilde{G}_1 Q} \right) + H\left( \frac{\tilde{F}_1}{F} \right) \right] d\lambda \\
& - \frac{\varphi_1 \tilde{F}_1 \tilde{K}_1 + \varphi_3 \tilde{F}_1 \tilde{G}_1}{\Lambda_2} \int_0^{\varrho_2} \bar{\Lambda}_2(\lambda) H\left( \frac{Q_\lambda \tilde{G}_1}{\tilde{Q}_1 G} \right) d\lambda \\
& - \frac{\varphi_1 \tilde{F}_1 \tilde{K}_1}{\Lambda_3} \int_0^{\varrho_3} \bar{\Lambda}_3(\lambda) H\left( \frac{G_\lambda \tilde{K}_1}{\tilde{G}_1 K} \right) d\lambda - \frac{\sigma \varphi_1 \tilde{F}_1 (\gamma + \varepsilon K)}{(\kappa + \sigma \tilde{U}_1)(\phi - \varepsilon \tilde{U}_1)} (U - \tilde{U}_1)^2.
\end{aligned}
$$

Hence, if $\tilde{\Re}_0 > 1$ then $\frac{d\tilde{\Theta}_1}{dt} \leq 0$ for all $F, Q, G, K, U > 0$. Also, $\frac{d\tilde{\Theta}_1}{dt} = 0$ when $F = \tilde{F}_1$, $Q = \tilde{Q}_1, G = \tilde{G}_1, K = \tilde{K}_1$ and $U = \tilde{U}_1$. Let $\bar{\Phi}_1'$ be the largest invariant subset of $\tilde{\Phi}_1 = \left\{ (F, Q, G, K, U) : \frac{d\tilde{\Theta}_1}{dt} = 0 \right\}$. Therefore, $\bar{\Phi}_1' = \{ \Xi \tilde{Q}_1 \}$. By applying L.I.P, we can conclude that if $\tilde{\Re}_0 > 1$, then the equilibrium $\Xi \tilde{Q}_1$ is G.A.S. $\quad \square$

## 4. Numerical Simulations

In this part, we accomplish some numerical simulations for both systems (5) and (23) to validate our theoretical results. Additionally, we investigate the impact of impaired humoral immunity on model (5), besides, the influence of time-delays on the dynamical system (23). We use ode45 and dde23 solvers in MATLAB to perform the numerical simulations for systems (5) and (23), respectively. Other methods can also be used for solving these systems (see e.g., [44,45]).

### 4.1. Numerical Simulation for Model (5)
4.1.1. Stability of Equilibria

In this part, we perform numerical simulations of system (5) using the parameter values provided in Table 1. Many of these parameter values are adopted from prior research. The remaining parameters, denoted as $\varphi_i$ for $i = 1, 2, 3$, are chosen specifically for the purpose of our numerical simulations. To assess the stability of the equilibria in system (5), we initiate the simulations with three distinct initial conditions, as outlined below:

**I.C.1:** $(F(0), Q(0), G(0), K(0), U(0)) = (400, 4, 2, 1, 0.4)$,
**I.C.2:** $(F(0), Q(0), G(0), K(0), U(0)) = (250, 5, 2.85, 3.5, 0.3)$,
**I.C.3:** $(F(0), Q(0), G(0), K(0), U(0)) = (500, 6.5, 4, 4, 0.1)$.

**Table 1.** Model parameters.

| Parameter | Value | Reference | Parameter | Value | Reference |
|---|---|---|---|---|---|
| $\psi$ | 10 | [46] | $\theta$ | 0.8 | [47] |
| $\rho$ | 0.01 | [46] | $\mu$ | 2.6 | [48] |
| $\varphi_1$ | varied | - | $\kappa$ | 2.4 | [48] |
| $\varphi_2$ | varied | - | $\sigma$ | 0.06 | [49] |
| $\varphi_3$ | varied | - | $\phi$ | 0.01 | [46] |
| $\zeta$ | 0.2 | [46] | $\gamma$ | 0.3 | [46] |
| $\vartheta$ | 0.17 | [46] | $\varepsilon$ | varied | - |

As $\Re_0$ is employed to regulate the stability of equilibria and is contingent on the infection rates $\varphi_i$ for $i = 1, 2, 3$, we introduce variations to these parameters. We then proceed to examine two distinct scenarios:

**Stability of $\Xi Q_0$.** We let $\varphi_1 = 0.0002$, $\varphi_2 = 0.0001$, $\varphi_3 = 0.0004$ and $\varepsilon = 0.001$. This gives $\Re_0 = 0.6869 < 1$. Figure 1 presents that the trajectories of the solution starting with I.C.1-I.C.3 end up at the equilibrium $\Xi Q_0 = (1000, 0, 0, 0, 0)$. In fact, this shows that $\Xi Q_0$ is G.A.S based to the result of Theorem 2. From a biological perspective, this scenario implies that the infection will be eradicated, and the human body will successfully eliminate the pathogen.

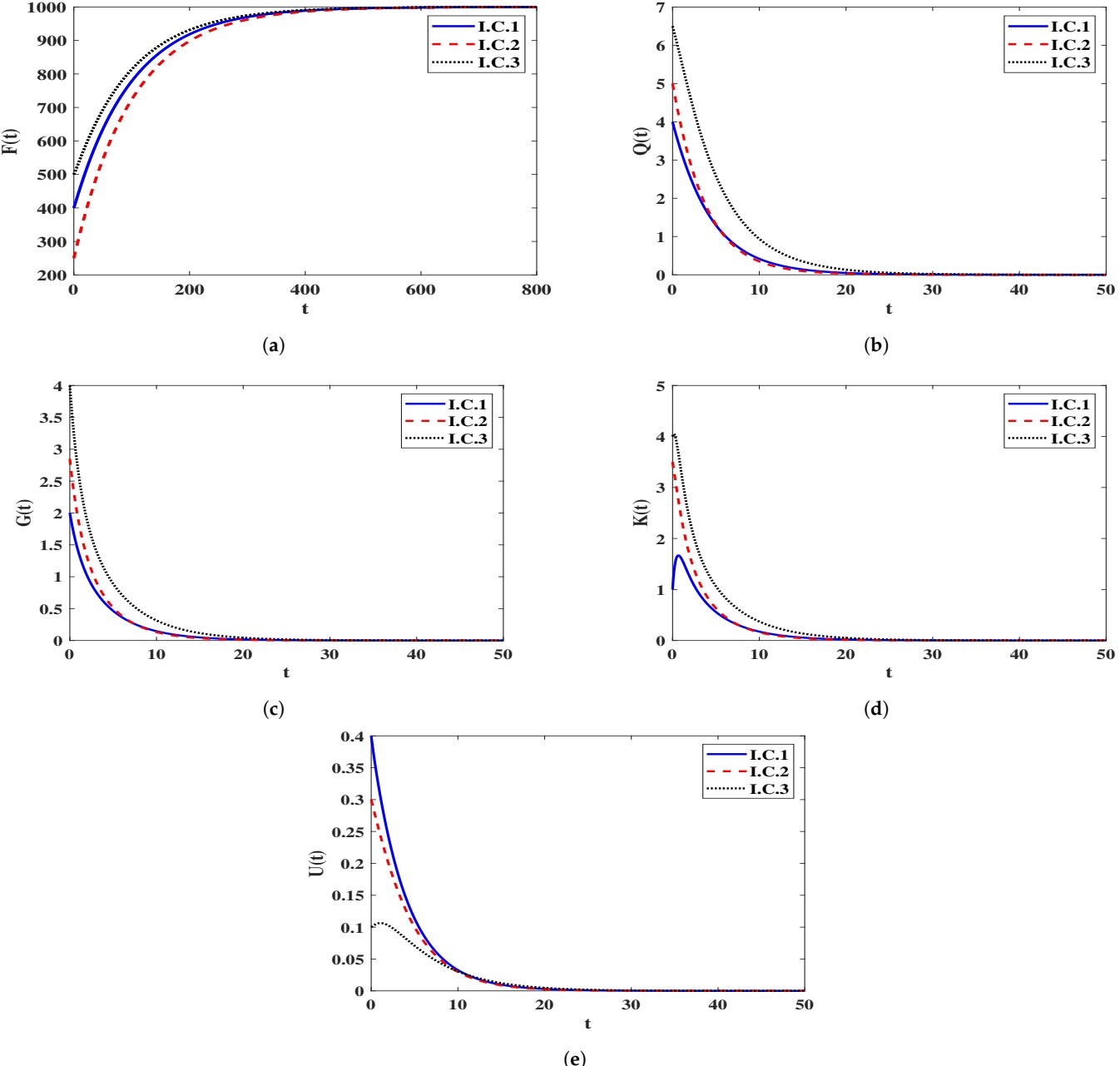

**Figure 1.** Solutions $(F(t), Q(t), G(t), K(t), U(t))$ of system (5) whenever $\Re_0 < 1$. (**a**) Healthy CD4$^+$T cells; (**b**) (HIV-1)-latently infected cells; (**c**) (HIV-1)-actively infected cells; (**d**) HIV-1 particles; (**e**) B-cells.

**Stability of $\Xi Q_1$.** We let $\varphi_1 = 0.003$, $\varphi_2 = 0.0001$, $\varphi_3 = 0.0004$ and $\varepsilon = 0.001$. With such choice we get $\Re_0 = 2.7365 > 1$. Clearly, the equilibrium point $\Xi Q_1$ exists when $\Re_0 > 1$ with $\Xi Q_1 = (366.543, 17.121, 4.28, 4.619, 0.152)$. Figure 2 demonstrates that the numerical outcomes come to an agreement with the result of Theorem 3 as the solutions of system (5) end up at $\Xi Q_1$ when $\Re_0 > 1$ for all I.C.1–I.C.3. From a biological standpoint, this situation reveals that both HIV-1 particles and B-cells will continue to exist within the host organism.

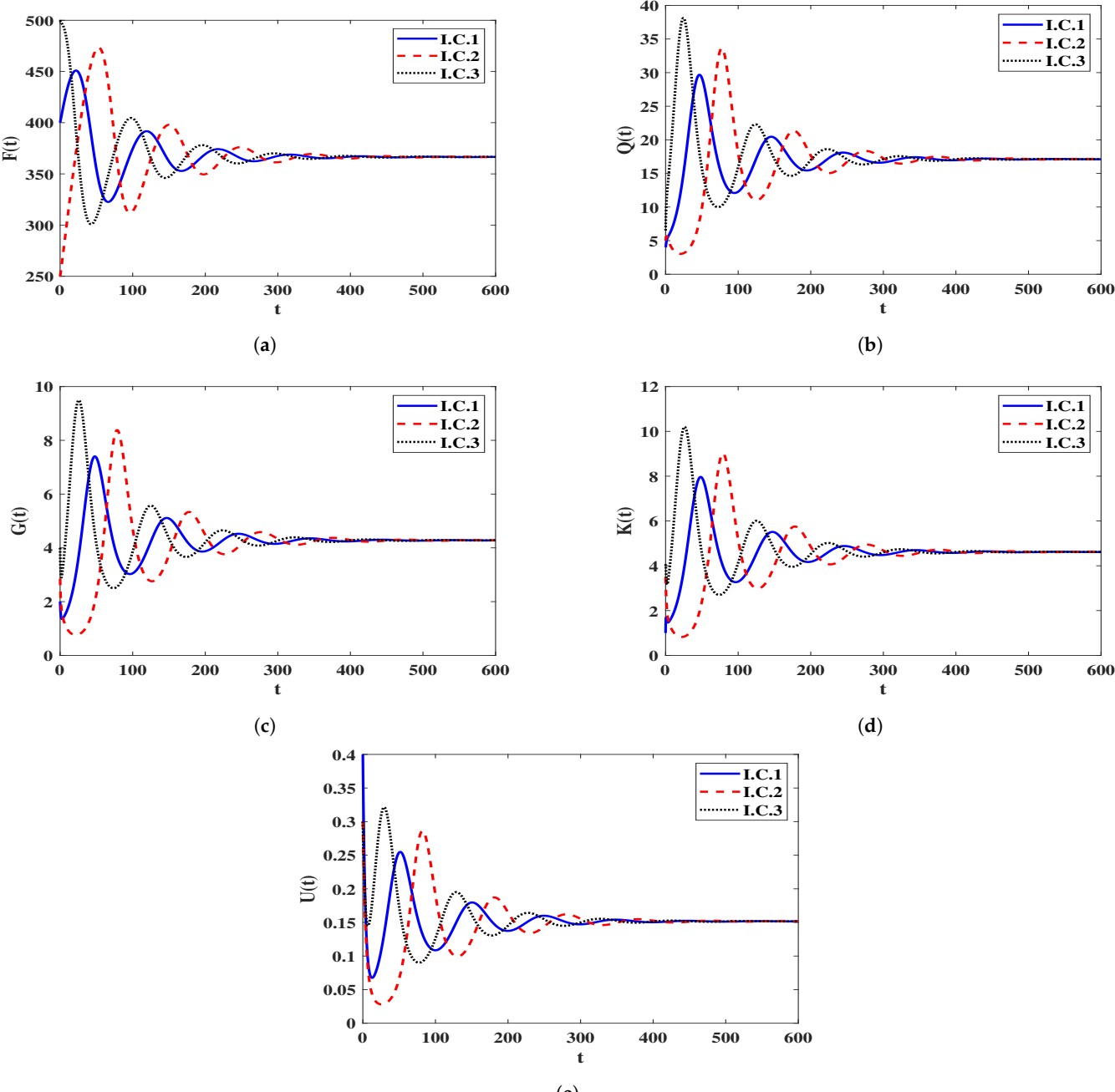

**Figure 2.** Solutions $(F(t), Q(t), G(t), K(t), U(t))$ of system (5) whenever $\Re_0 > 1$. (**a**) Healthy CD4$^+$T cells; (**b**) (HIV-1)-latently infected cells; (**c**) (HIV-1)-actively infected cells; (**d**) HIV-1 particles; (**e**) B-cells.

### 4.1.2. Effect of the Impaired Humoral Immunity

In this scenario, we introduce variations in the parameter $\varepsilon$ while setting specific values for $\varphi_1 = 0.003$, $\varphi_2 = 0.0001$, and $\varphi_3 = 0.0004$. To explore how the dynamics of system (5) are influenced by the impairment of the humoral immune response, we numerically solve the system, considering different values of $\varepsilon$ as outlined in Table 2. In this case, we define the following initial condition:

**I.C.4:** $(F(0), Q(0), G(0), K(0), U(0)) = (366, 17.15, 4.28, 4.6, 0.08)$.

**Table 2.** Effect of the impaired humoral immunity parameter.

| $\varepsilon$ | Equilibria |
|---|---|
| $\varepsilon = 0$ | $\Xi Q_1 = (366.56, 17.12, 4.28, 4.619, 0.154)$ |
| $\varepsilon = 0.04$ | $\Xi Q_1 = (366.131, 17.132, 4.283, 4.629, 0.095)$ |
| $\varepsilon = 0.1$ | $\Xi Q_1 = (365.877, 17.139, 4.285, 4.635, 0.061)$ |
| $\varepsilon = 2$ | $\Xi Q_1 = (365.468, 17.15, 4.287, 4.644, 0.005)$ |

We observe from Table 2 that an increase in $\varepsilon$ results in a decrease in the number of B-cells. This reduction is linked to a higher count of (HIV-1)-latently and (HIV-1)-actively infected cells, as well as HIV-1 particles. As a consequence, the population of healthy CD4$^+$T cells decreases. An insightful observation from Figure 3 is that the impairment of the humoral immune response does not alter the stability criteria of the equilibria. This is evident as the parameter $\Re_0$ remains unaffected by changes in $\varepsilon$.

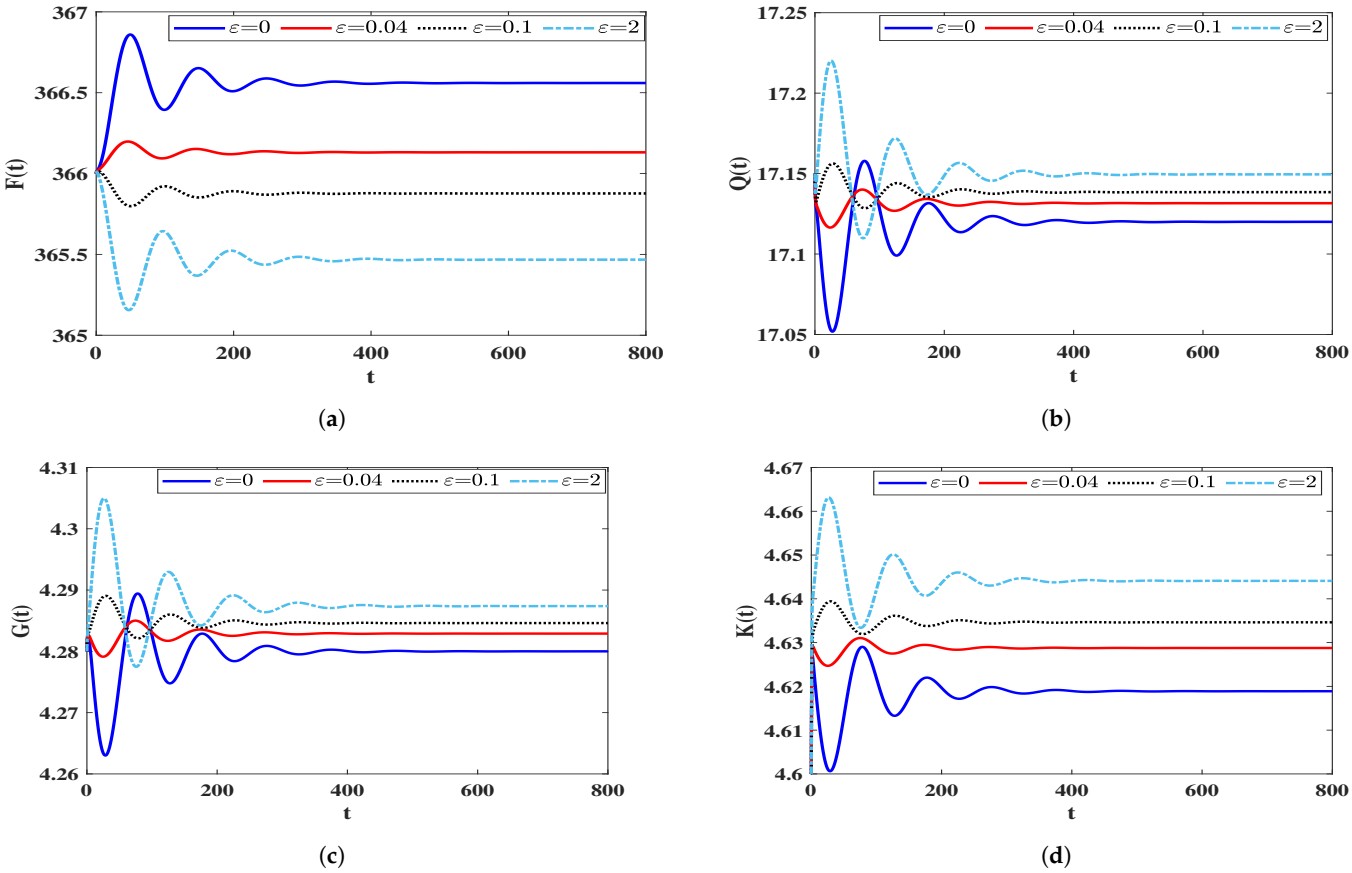

**Figure 3.** *Cont.*

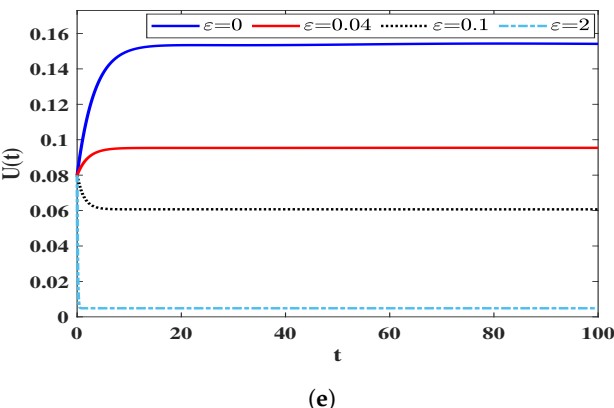

(**e**)

**Figure 3.** Solutions of system (5) for various $\varepsilon$ values. (**a**) Healthy CD4$^+$T cells; (**b**) (HIV-1)-latently infected cells; (**c**) (HIV-1)-actively infected cells; (**d**) HIV-1 particles; (**e**) B-ells.

*4.2. Numerical Simulation for Model (23)*

In this subsection, for numerical computations, we choose a specific form for the probability distribution functions $T_i(\lambda)$, where $i = 1, 2, 3$, as outlined below:

$$T_i(\lambda) = \delta_*(\lambda - \lambda_i), \quad \lambda_i \in [0, \varrho_i], \quad i = 1, 2, 3,$$

where $\delta_*(.)$ is the Dirac delta function. As $\varrho_i \to \infty$, we obtain

$$\int_0^\infty T_i(\lambda)d\lambda = 1, \quad i = 1, 2, 3.$$

Moreover, we have

$$\Lambda_i = \int_0^\infty \delta_*(\lambda - \lambda_i)e^{-\alpha_i\lambda}d\lambda = e^{-\alpha_i\lambda_i}, \quad i = 1, 2, 3.$$

Therefore, the distributed-time delay system (23) will be converted into a discrete-time delay system as follows:

$$
\begin{cases}
\frac{dF(t)}{dt} = \psi - \rho F(t) - \varphi_1 F(t)K(t) - \varphi_2 F(t)Q(t) - \varphi_3 F(t)G(t), \\
\frac{dQ(t)}{dt} = e^{-\alpha_1\lambda_1}F(t - \lambda_1)(\varphi_1 K(t - \lambda_1) + \varphi_2 Q(t - \lambda_1) \\
\qquad\quad + \varphi_3 G(t - \lambda_1)) - (\xi + \vartheta)Q(t), \\
\frac{dG(t)}{dt} = \xi e^{-\alpha_2\lambda_2}Q(t - \lambda_2) - \theta G(t), \\
\frac{dK(t)}{dt} = \mu e^{-\alpha_3\lambda_3}G(t - \lambda_3) - \kappa K(t) - \sigma U(t)K(t), \\
\frac{dU(t)}{dt} = \phi K(t) - \gamma U(t) - \varepsilon U(t)K(t).
\end{cases}
\tag{40}
$$

For system (40), the basic reproductive number is:

$$\tilde{\mathfrak{R}}_{0(40)} = \frac{\tilde{F}_0 e^{-\alpha_1\lambda_1}\left(\xi e^{-\alpha_2\lambda_2}\left(\varphi_1 \mu e^{-\alpha_3\lambda_3} + \varphi_3\kappa\right) + \varphi_2\kappa\theta\right)}{(\xi + \vartheta)\kappa\theta}. \tag{41}$$

The Effect of the Time-Delays on the Stability of Equilibria

To explore how the solutions of the system are affected by the time-delay parameters (40), we keep the parameters constant $\varphi_1 = 0.003$, $\varphi_2 = 0.0001$, $\varphi_3 = 0.0004$, $\varepsilon = 0.001$, $\alpha_1 = 0.1$, $\alpha_2 = 0.2$ and $\alpha_3 = 0.3$. On the contrary, the remaining parameters will be selected from Table 1. Furthermore, we will vary the delay parameters $\lambda_i$, where $i = 1, 2, 3$. The dependence of $\tilde{\mathfrak{R}}_{0(40)}$ which is presented in Equation (41) on the values of $\lambda_i$ causes a remarkably changing in the stability of equilibria as long as parameters $\lambda_i$ are changed. Let's consider the following scenarios for the delay values:

**DV1** $\lambda_1 = 0.07$, $\lambda_2 = 0.06$, $\lambda_3 = 0.05$.
**DV2** $\lambda_1 = 0.8$, $\lambda_2 = 0.7$, $\lambda_3 = 0.9$.
**DV3** $\lambda_1 = 1.3$, $\lambda_2 = 1.4$, $\lambda_3 = 1.5$.
**DV4** $\lambda_1 = 1.8$, $\lambda_2 = 1.9$, $\lambda_3 = 2$.
**DV5** $\lambda_1 = 4$, $\lambda_2 = 3$, $\lambda_3 = 5$.

We solve system (40) under the below initial

**I.C.5:** $(F(r), Q(r), G(r), K(r), U(r)) = (400, 4, 2, 1, 0.3)$, $r \in [-\lambda, 0]$, $\lambda = \max\{\lambda_1, \lambda_2, \lambda_3\}$.

In Table 3, we present the values of $\tilde{\mathfrak{R}}_{0(40)}$ for different values of $\lambda_i$, $i = 1, 2, 3$. We notice that an increase in the parameters $\lambda_i$ leads to a remarkable decrease in the values of $\tilde{\mathfrak{R}}_{0(40)}$. Figure 4 illustrates the numerical solutions of the system. A significant effect of the inclusion of time-delays is concluded, that is an increase in the number of healthy CD4$^+$T cells and a decrease in the numbers of (HIV-1)-latently and (HIV-1)-actively infected cells, HIV-1 particles, and B-cells occur.

**Table 3.** The disparity of $\tilde{\mathfrak{R}}_{0(40)}$ based on $\lambda_i$.

| Delay Parameters $(\lambda_1, \lambda_2, \lambda_3)$ | Equilibria | $\tilde{\mathfrak{R}}_{0(40)}$ |
|---|---|---|
| $(0.07, 0.06, 0.05)$ | $\Xi\tilde{Q}_{1(40)} = (377.574, 16.705, 4.126, 4.388, 0.144)$ | $\tilde{\mathfrak{R}}_{0(40)} = 2.656$ |
| $(0.8, 0.7, 0.9)$ | $\Xi\tilde{Q}_{1(40)} = (552.652, 11.161, 2.426, 2.003, 0.066)$ | $\tilde{\mathfrak{R}}_{0(40)} = 1.812$ |
| $(1.3, 1.4, 1.5)$ | $\Xi\tilde{Q}_{1(40)} = (743.32, 6.092, 1.151, 0.795, 0.026)$ | $\tilde{\mathfrak{R}}_{0(40)} = 1.346$ |
| $(1.8, 1.9, 2)$ | $\Xi\tilde{Q}_{1(40)} = (935.941, 1.446, 0.247, 0.147, 0.005)$ | $\tilde{\mathfrak{R}}_{0(40)} = 1.069$ |
| $(4, 3, 5)$ | $\Xi\tilde{Q}_{0(40)} = (1000, 0, 0, 0, 0)$ | $\tilde{\mathfrak{R}}_{0(40)} = 0.461$ |

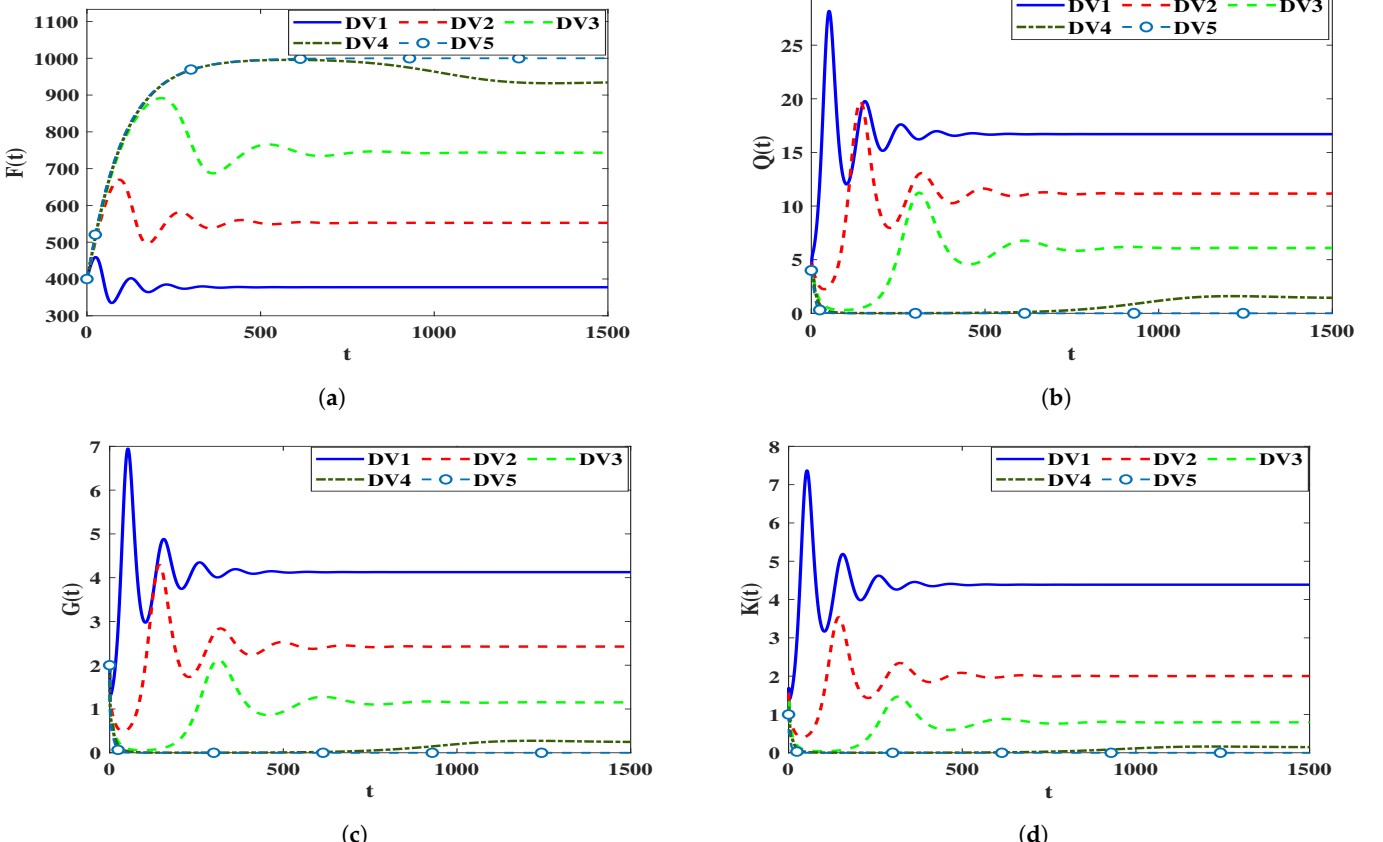

(**a**)

(**b**)

(**c**)

(**d**)

**Figure 4.** *Cont.*

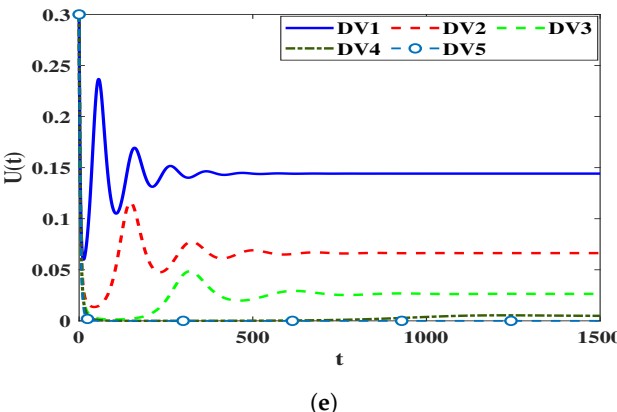

(**e**)

**Figure 4.** The influence of the time-delay parameters $\lambda_i$ on the solutions of the system (40). (**a**) Healthy CD4$^+$T cells; (**b**) (HIV-1)-latently infected cells; (**c**) (HIV-1)-actively infected cells; (**d**) HIV-1 particles; (**e**) B-cells.

*4.3. Sensitivity Analysis*

4.3.1. Sensitivity Analysis for Model (5)

The primary objective of a sensitivity analysis is to pinpoint the variable with the highest risk contribution. To achieve this, we will employ partial derivatives to compute sensitivity indices when variables undergo variations based on parameters. The normalized forward sensitivity index for the basic reproductive number, $\Re_0$, can be expressed in terms of parameters as follows:

$$\Xi_S = \frac{S}{\Re_0} \frac{\partial \Re_0}{\partial S},\qquad(42)$$

where $S$ is a given parameter. We used Equation (42) to determine the sensitivity indices for each parameter contained in the basic reproductive number, $\Re_0$, using the parameter values provided in Table 1, as well as the following parameters: $\varphi_1 = 0.003$, $\varphi_2 = 0.0001$ and $\varphi_3 = 0.0004$. Table 4 and Figure 5 present the sensitivity index values for $\Re_0$. It is evident that $\psi$, $\varphi_1$, $\varphi_2$, $\varphi_3$, $\mu$ and $\xi$ have positive indices values. In this instance, there is a correlation between the endemicity of the HIV-1 disease and an increase in the values of these parameters. The other indices are negative, which means that when the values of $\rho$, $\kappa$, $\vartheta$ and $\theta$ increase, the value of the basic reproductive number, $\Re_0$, decreases. Obviously, the most crucial parameters in terms of sensitivity are $\psi$, $\varphi_1$ and $\mu$, while $\varphi_2$ and $\varphi_3$ are the least crucial. The parameter of B-cells responsiveness, $\phi$, has no effect on $\Re_0$.

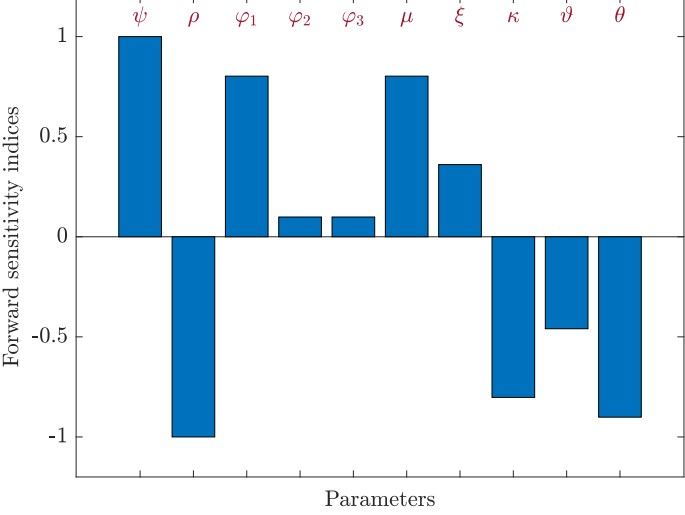

**Figure 5.** Forward sensitivity analysis of the parameters on $\Re_0$ in system (5).

**Table 4.** Sensitivity index of $\Re_0$ of model (5).

| Parameter $S$ | Value of $\Xi_S$ | Parameter $S$ | Value of $\Xi_S$ |
|---|---|---|---|
| $\psi$ | 1 | $\mu$ | 0.802 |
| $\rho$ | $-1$ | $\zeta$ | 0.361 |
| $\varphi_1$ | 0.802 | $\kappa$ | $-0.802$ |
| $\varphi_2$ | 0.099 | $\vartheta$ | $-0.459$ |
| $\varphi_3$ | 0.099 | $\theta$ | $-0.901$ |

4.3.2. Sensitivity Analysis for Model (40)

We applied Equation (42) with respect to $\tilde{\Re}_{0(40)}$ to compute the sensitivity indices for each parameter contained in the basic reproductive number, $\tilde{\Re}_{0(40)}$, using the parameter values provided in Table 1, as well as the following parameters: $\varphi_1 = 0.003$, $\varphi_2 = 0.0001$, $\varphi_3 = 0.0004$, $\alpha_1 = 0.1$, $\alpha_2 = 0.2$, $\alpha_3 = 0.3$, $\lambda_1 = 0.07$, $\lambda_2 = 0.06$, and $\lambda_3 = 0.05$. Table 5 and Figure 6 present the sensitivity index values for $\tilde{\Re}_{0(40)}$. Since, $\psi$, $\varphi_1$, $\varphi_2$, $\varphi_3$, $\mu$ and $\zeta$ have positive indices, then, increasing these values will increase endemic of the HIV-1 disease. While increasing negative indices values, which are $\kappa$, $\vartheta$, $\theta$, $\alpha_1$, $\alpha_2$, $\alpha_3$, $\lambda_1$, $\lambda_2$, and $\lambda_3$ will decrease the value of $\tilde{\Re}_{0(40)}$. We can see that $\psi$, $\varphi_1$ and $\mu$ are the most important parameters, and $\varphi_2$ and $\varphi_3$ are the least important. The parameter of B-cells responsiveness, $\phi$, has no effect on $\tilde{\Re}_{0(40)}$.

**Table 5.** Sensitivity index of $\tilde{\Re}_{0(40)}$ of model (40).

| Parameter $S$ | Value of $\Xi_S$ | Parameter $S$ | Value of $\Xi_S$ |
|---|---|---|---|
| $\psi$ | 1 | $\vartheta$ | $-0.459$ |
| $\rho$ | $-1$ | $\theta$ | $-0.899$ |
| $\varphi_1$ | 0.799 | $\alpha_1$ | $-0.007$ |
| $\varphi_2$ | 0.101 | $\alpha_2$ | $-0.011$ |
| $\varphi_3$ | 0.1 | $\alpha_3$ | $-0.012$ |
| $\mu$ | 0.8 | $\lambda_1$ | $-0.007$ |
| $\zeta$ | 0.358 | $\lambda_2$ | $-0.011$ |
| $\kappa$ | $-0.799$ | $\lambda_3$ | $-0.012$ |

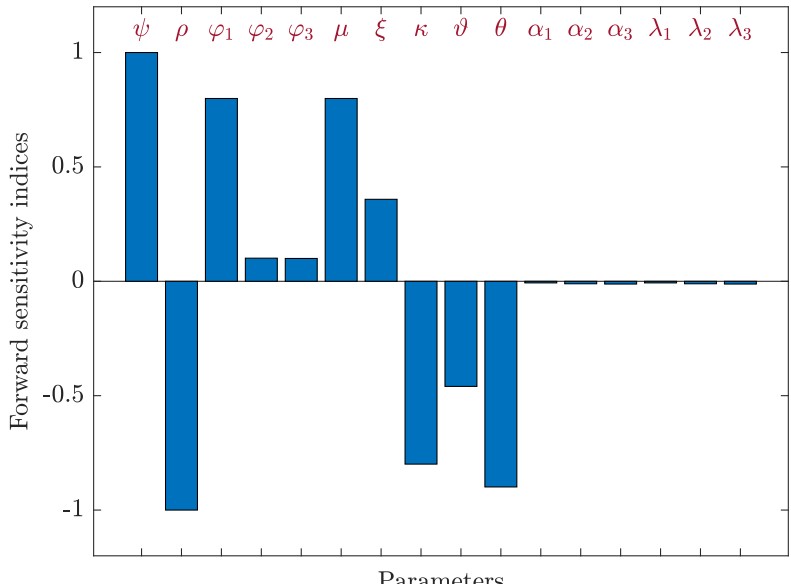

**Figure 6.** Forward sensitivity analysis of the parameters on $\tilde{\Re}_{0(40)}$ of system (40).

## 5. Discussion

To show the importance of including the latent CI spread in our proposed models, we consider model (5) under the effect of three types of antiviral drug therapies as:

$$
\begin{cases}
\frac{dF(t)}{dt} &= \psi - \rho F(t) - (1-\ell_1)\varphi_1 F(t)K(t) - (1-\ell_2)\varphi_2 F(t)Q(t) - (1-\ell_3)\varphi_3 F(t)G(t), \\
\frac{dQ(t)}{dt} &= (1-\ell_1)\varphi_1 F(t)K(t) + (1-\ell_2)\varphi_2 F(t)Q(t) + (1-\ell_3)\varphi_3 F(t)G(t) - (\xi + \vartheta)Q(t), \\
\frac{dG(t)}{dt} &= \xi Q(t) - \theta G(t), \\
\frac{dK(t)}{dt} &= \mu G(t) - \kappa K(t) - \sigma U(t)K(t), \\
\frac{dU(t)}{dt} &= \phi K(t) - \gamma U(t) - \varepsilon U(t)K(t),
\end{cases}
\tag{43}
$$

where $\ell_1 \in [0,1]$ is the efficacy of antiviral therapy in blocking VI. Moreover, $\ell_2 \in [0,1]$ and $\ell_3 \in [0,1]$ are the efficacies of therapy in blocking latent CI and active CI, respectively [50]. The basic reproductive number of system (43) is:

$$
\Re_0 = \frac{(1-\ell_1)F_0\mu\xi\varphi_1}{\kappa\theta(\xi + \vartheta)} + \frac{(1-\ell_2)F_0\varphi_2}{\xi + \vartheta} + \frac{(1-\ell_3)F_0\xi\varphi_3}{\theta(\xi + \vartheta)}.
$$

We assume that $\ell = \ell_1 = \ell_2 = \ell_3$, then we get

$$
\Re_0^\ell = (1-\ell)\left[\frac{F_0\mu\xi\varphi_1}{\kappa\theta(\xi + \vartheta)} + \frac{F_0\varphi_2}{\xi + \vartheta} + \frac{F_0\xi\varphi_3}{\theta(\xi + \vartheta)}\right] = (1-\ell)\Re_0.
$$

Now, we calculate the drug efficacy $\ell$ that makes $\Re_0^\ell < 1$ and stabilizes $\Xi Q_0$ of system (43) as:

$$
1 \geq \ell > \tilde{\ell}_{\min} = \max\left\{0, 1 - \frac{1}{\Re_0}\right\}.
\tag{44}
$$

When we ignore the latent CI spread in model (43) we obtain

$$
\begin{cases}
\frac{dF(t)}{dt} &= \psi - \rho F(t) - (1-\ell)\varphi_1 F(t)K(t) - (1-\ell)\varphi_3 F(t)G(t), \\
\frac{dQ(t)}{dt} &= (1-\ell)\varphi_1 F(t)K(t) + (1-\ell)\varphi_3 F(t)G(t) - (\xi + \vartheta)Q(t), \\
\frac{dG(t)}{dt} &= \xi Q(t) - \theta G(t), \\
\frac{dK(t)}{dt} &= \mu G(t) - \kappa K(t) - \sigma U(t)K(t), \\
\frac{dU(t)}{dt} &= \phi K(t) - \gamma U(t) - \varepsilon U(t)K(t),
\end{cases}
\tag{45}
$$

and the basic reproductive number of model (45) is:

$$
\hat{\Re}_0^\ell = (1-\ell)\left[\frac{F_0\mu\xi\varphi_1}{\kappa\theta(\xi + \vartheta)} + \frac{F_0\xi\varphi_3}{\theta(\xi + \vartheta)}\right] = (1-\ell)\hat{\Re}_0.
$$

We determine the drug efficacy $\ell$ that makes $\hat{\Re}_0^\ell < 1$ and stabilizes $\Xi Q_0$ of system (45) as:

$$
1 \geq \ell > \hat{\ell}_{\min} = \max\left\{0, 1 - \frac{1}{\hat{\Re}_0}\right\}.
\tag{46}
$$

Clearly, $\hat{\Re}_0 < \Re_0$, then comparing Equations (44) and (46) we get that $\hat{\ell}_{\min} \leq \tilde{\ell}_{\min}$. Therefore, if we apply drugs with efficacy $\ell$ such that $\hat{\ell}_{\min} \leq \ell < \tilde{\ell}_{\min}$, this guarantees that $\hat{\Re}_0^\ell < 1$ and then $\Xi Q_0$ of system (45) is G.A.S, however, $\Re_0^\ell > 1$ and then $\Xi Q_0$ of system (43) is unstable. Consequently, the designed drug therapies using a model without considering the latent CI spread may be inaccurate or insufficient to eradicate the viruses from the body. Therefore, our proposed models are more relevant in describing the HIV-1 dynamics than the models presented in [29,30].

The primary limitation of our current study is that we did not utilize real data to estimate the model's parameter values. Several factors contribute to this limitation:

1.  Limited Availability of Real Data: There is a scarcity of real data from HIV-1 infected individuals, which hinders the accurate estimation of model parameters.
2.  Precision Issues: Comparing our obtained results with the limited existing studies may lack precision due to the scarcity of data points.
3.  Data Collection Challenges: Collecting real data from HIV-1 infected patients can be a challenging and resource-intensive task.
4.  Experimental Scope: Conducting experiments to obtain real data falls outside the scope of this paper.

Therefore, it is crucial to acknowledge that the theoretical findings presented in this paper should be validated against empirical observations when sufficient real data becomes accessible.

## 6. Conclusions

In this study, we developed two models to get an insight into HIV-1 dynamics taking impaired humoral immunity under consideration. These models consist of five compartments; healthy $CD4^{+}T$ cells, (HIV-1)-latently infected cells, (HIV-1)-actively infected cells, free HIV-1 particles, and B-cells. In pursuit of a more realistic representation, we considered a scenario where healthy $CD4^{+}T$ cells become susceptible to infection upon encountering free HIV-1 particles, (HIV-1)-latently infected cells, and (HIV-1)-actively infected cells. In our second model, we introduced three distributed time-delays to better capture the dynamics. It is noteworthy that the solutions generated by these models exhibit nonnegative and bounded characteristics. Within this framework, we identified two critical equilibria: the infection-free equilibrium denoted as $\Xi Q_0$ (or $\Xi \tilde{Q}_0$) and the infected equilibrium represented as $\Xi Q_1$ (or $\Xi \tilde{Q}_1$). To quantify the impact and potential outcomes, we computed the basic reproductive numbers, denoted as $\Re_0$ (or $\tilde{\Re}_0$). These values play a pivotal role in dictating the existence and global stability of the aforementioned equilibria. Notably, $\Re_0$ (or $\tilde{\Re}_0$) comprises three distinct components: the contribution from viral infection (VI), the contribution arising from latent cellular infection (latent CI), and the contribution attributed to active cellular infection (active CI). To assess the overall system behavior, we employed Lyapunov functions and the LaSalle's invariance principle (L.I.P) to investigate the global asymptotic stability of these equilibria. Our analysis yielded two important scenarios: first, if $\Re_0 < 1$ (or $\tilde{\Re}_0 < 1$), then the infection-free equilibrium $\Xi Q_0$ (or $\Xi \tilde{Q}_0$) is globally asymptotically stable (G.A.S), leading to eventual infection extinction. Conversely, if $\Re_0 > 1$ (or $\tilde{\Re}_0 > 1$), the equilibrium $\Xi Q_0$ (or $\Xi \tilde{Q}_0$) becomes unstable, and the infected equilibrium $\Xi Q_1$ (or $\Xi \tilde{Q}_1$) prevails as G.A.S, signifying the establishment of chronic infection. To reinforce our theoretical findings, we conducted numerical simulations that corroborated our analytical results. Furthermore, we delved into the impact of B-cell impairment, time-delay, and latent CI on the dynamics of HIV-1. Notably, weakened immunity emerged as a significant contributor to disease progression. Additionally, the presence of time-delay emerged as a key factor in reducing the basic reproductive number $\tilde{\Re}_0$ and consequently suppressing HIV-1 replication. In light of this, strategies aimed at eliminating HIV-1 from the body should prioritize measures that reduce $\tilde{\Re}_0$ below 1. We also observed an increase in delay parameters $\lambda_i$, where $i = 1, 2, 3$, when infected patients undergo drug therapies against HIV-1. In a crucial finding, we highlighted the consequences of neglecting latent CI spread within the HIV-1 dynamics model. This omission can lead to an underestimation of the basic reproductive number, potentially resulting in inaccurate or insufficient drug dosing for virus eradication. This underscores the vital importance of incorporating latent CI spread within our proposed models. In addition, we conducted a sensitivity analysis to elucidate how variations in the values of all model parameters can impact $\Re_0$ (or $\tilde{\Re}_0$) under specific data conditions. This comprehensive analysis provided valuable insights into the sensitivity of the system to parameter changes, further enhancing our understanding of HIV-1 dynamics.

*Future Works*

The following enhancements can be made to extend Model (23):

- Including the diffusion of the cells and viruses as [8,9]:

$$\frac{\partial F(t,h)}{\partial t} - \phi_F \Delta F(t,h) = \psi - \rho F(t,h) - \varphi_1 F(t,h)K(t,h) - \varphi_2 F(t,h)Q(t,h) - \varphi_3 F(t,h)G(t,h),$$

$$\frac{\partial Q(t,h)}{\partial t} - \phi_Q \Delta Q(t,h) = \int_0^{\varrho_1} T_1(\lambda)e^{-\alpha_1\lambda}F(t-\lambda,h)(\varphi_1 K(t-\lambda,h) + \varphi_2 Q(t-\lambda,h)$$
$$+ \varphi_3 G(t-\lambda,h))d\lambda - (\xi + \vartheta)Q(t,h),$$

$$\frac{\partial G(t,h)}{\partial t} - \phi_G \Delta G(t,h) = \xi \int_0^{\varrho_2} T_2(\lambda)e^{-\alpha_2\lambda}Q(t-\lambda,h)d\lambda - \theta G(t,h),$$

$$\frac{\partial K(t,h)}{\partial t} - \phi_K \Delta K(t,h) = \mu \int_0^{\varrho_3} T_3(\lambda)e^{-\alpha_3\lambda}G(t-\lambda,h)d\lambda - \kappa K(t,h) - \sigma U(t,h)K(t,h),$$

$$\frac{\partial U(t,h)}{\partial t} - \phi_U \Delta U(t,h) = \phi K(t,h) - \gamma U(t,h) - \varepsilon U(t,h)K(t,h),$$

where $h$ is the position, $\Delta = \frac{\partial^2}{\partial h^2}$ and $\phi_u$ is the diffusion coefficient of compartment $u$. One can also include different kinds of diffusion in our systems (see e.g., [51–53]).
- Utilizing real-world data to estimate model parameters accurately, which can enhance the model's predictive capabilities and align it better with empirical observations.
- Broadening the scope of the model to incorporate the role of Cytotoxic T Lymphocytes (CTLs) alongside B-cells, allowing for a more comprehensive representation of the immune response.
- Investigating the integration of age structure into the infected cell population within the model, which can provide insights into how age-related factors impact disease dynamics.
- Exploring the effects of viral mutations on the dynamics of the model, considering how genetic changes in the virus may influence disease progression and response to interventions.

It should be noted that these proposed enhancements are being deferred for future consideration and further study.

**Author Contributions:** Conceptualization, N.H.A. and A.M.E.; Methodology, N.H.A.; Software, N.H.A., R.H.H. and A.M.E.; Validation, N.H.A. and A.M.E.; Formal analysis, N.H.A. and R.H.H.; Investigation, N.H.A.; Resources, N.H.A. and A.M.E.; Writing—original draft, N.H.A., R.H.H. and A.M.E.; Writing—review & editing, W.S.; Visualization, N.H.A. and A.M.E.; Supervision, N.H.A. and A.M.E. All authors have read and agreed to the published version of the manuscript.

**Funding:** This work was funded by the University of Jeddah, Jeddah, Saudi Arabia, under grant No. (UJ-22-DR-102).

**Data Availability Statement:** Not applicable.

**Acknowledgments:** This work was funded by the University of Jeddah, Jeddah, Saudi Arabia, under grant No. (UJ-22-DR-102). The authors, therefore, acknowledge with thanks the University of Jeddah for its technical and financial support.

**Conflicts of Interest:** The authors declare that they have no conflict of interest regarding the publication of this paper.

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
