# Peer review of "Stability of Impaired Humoral Immunity HIV-1 Models with Active and Latent Cellular Infections"

_computation, doi:10.3390/computation11100207_

Round 1

Reviewer 1 Report

The described work seems to be very interesting and important from a scientific point of view. It is focused on mathematical modeling of the dynamics of the HIV-1 virus in human organisms, especially in the context of weakened humoral immunity. The authors of the paper constructed and analyzed two mathematical models describing the interaction between HIV-1 and the immune system. They introduced several key compartments that reflect different populations of cells and molecules associated with infection. The work examined whether the model solutions remain non-negative and limited, which is important from the point of view of biological sense, the basic reproductive number was calculated, which is an important parameter in epidemiology, determining the ability of the virus to spread in the population, all possible equilibrium points in the model were identified and their global stability was studied, which allows to understand the long-term behavior of the system, numerical simulations were performed that confirm the theoretical results and allow for a better understanding of how variable parameters affect HIV-1 dynamics, the effect of B-cell attenuation and the effect of latency time on HIV-1 dynamics were investigated 1, which helps to better understand how these factors can affect the course of infection.

The work is well written, although the authors did not use the appropriate template, also the use of LaTeX would improve the quality (editing part) of the work.

The analysis of equilibria and their global stability using mathematical tools such as Lyapunov's method and LaSalle's principle of invariance is interesting. Demonstrating the conditions under which an infection resolves or becomes chronic provides a deeper understanding of infection dynamics. Numerical simulations confirming theoretical results add credibility and value to your work. The final conclusion that weakened immunity has a significant impact on disease development and that time delays may be crucial in shaping treatment strategies is an important contribution to the field of antiviral therapies. The conclusions contained in the work may have a significant impact on improving the understanding of the dynamics of HIV-1 in human organisms and on the development of more effective therapeutic strategies. The work is perfectly written and the analysis is well conducted, which proves the diligence and commitment of the authors.

Of course, as in any work, there are also a few caveats in this one:

1st edition mentioned above - the work is not written in the appropriate template, I also encourage authors to use the LaTeX editor

2. although the described models are complex and contain many compartments, the paper does not provide a sufficiently clear explanation of the important assumptions and parameters of the model. Readers lack a deeper understanding of why the selected cells and molecules were included, and what the mathematical basis of these choices is

3. although the authors discuss numerical results, there is no detailed comparison of these results with available empirical data or other studies. This limits the power of conclusions and their practical application. It would also be useful to discuss potential sources of numerical errors or assumptions that may affect the accuracy of the results

3. The paper did not sufficiently develop the topic of how specific results might affect therapeutic strategies and treatment of HIV-1

4. the paper could use more references to existing research and scientific papers to reinforce and place its findings in a broader context

I also found some syntax slips, I didn't mark most of them, I chose only a few typical ones, which I marked in the pdf file.

Author Response

Response to Reviewers
Stability of impaired humoral immunity HIV-1 models with active and latent cellular infections

Noura H. AlShamrani , Reham H. Halawani, Wafa Shammakh and Ahmed M. Elaiw

    First we thank the Editor who handling our paper. We also would like to thank the anonymous reviewers for their valuable comments and suggestions. We have revised the paper carefully in accordance with these comments and suggestions. The changes in the paper are marked by "bold face".

Replies to the comments of Reviewer #1

    First we would like to thank the reviewer for reading our paper.
    1st edition mentioned above - the work is not written in the appropriate template, I also encourage authors to use the LaTeX editor
    Reply: We have used LaTeX editor. Once the paper is accepted, we will use the template of the journal.
    2. although the described models are complex and contain many compartments, the paper does not provide a sufficiently clear explanation of the important assumptions and parameters of the model. Readers lack a deeper understanding of why the selected cells and molecules were included, and what the mathematical basis of these choices is
    Reply: We have defind all variables and paramters given in the models. We have also explained the interactions between the HIV-1 particles, target cells and immune cells.
    3. although the authors discuss numerical results, there is no detailed comparison of these results with available empirical data or other studies. This limits the power of conclusions and their practical application. It would also be useful to discuss potential sources of numerical errors or assumptions that may affect the accuracy of the results
    Reply: In the discussion section, we have added a comparison between our proposed model and the models presented in the literature. Moreover, we have added the following:
    "The main limitation of of our present work is that we did not use real data to estimate the values of the model's parameters. The main reasons are as follows: (1) The real data from HIV-1 infected individuals are still few; (2) Comparing our obtained results with a few number of real studies may not be very precise; (3) Collecting real data from HIV-1 infected patients is not an easy task; (4) Working on experiments to obtain real data is beyond the scope of this paper. Thus, the theoretical results obtained in this paper need to be tested against empirical findings when real data become available."
    4. The paper did not sufficiently develop the topic of how specific results might affect therapeutic strategies and treatment of HIV-1.
    Reply: In discussion section we have added some discussions about the effect of latent CI on designing the antiviral treatment. 
    4. the paper could use more references to existing research and scientific papers to reinforce and place its findings in a broader context.
    Reply: Some important references have been added.
    I also found some syntax slips, I didn't mark most of them, I chose only a few typical ones, which I marked in the pdf file.
    Reply: We haved fixed them.

Replies to the comments of Reviewer #2

    First we would like to thank the reviewer for reading our paper and for his constructive and valuable comments.
    This is an interesting manuscript dealing with two different models of within-host dynamics of human immunodeciency virus type-1 (HIV-1) with impaired humoral immunity. The results are important in uderstanding HIV disease and its dynamics. It would be better if the authors could give physical/biological interpretation of the basic reproduction number and discuss about the implication in clinical practice. Moreover, sensitivity analysis is also needed to show the significance both the parameters, especially those appeared in the basic reproductio number.
    Reply: We have added the biological interpretation of the basic reproduction number and discussed about the implication in clinical practice. Moreover, we have added a subsection for sensitivity analysis.

Replies to the comments of Reviewer #3

    First we would like to thank the reviewer for reading our paper and for his constructive and valuable comments.
    I have carefully reviewed the manuscript titled "Stability of Impaired Humoral Immunity HIV-1 Models with Active and Latent Cellular Infections." I appreciate the authors' efforts in exploring the within-host dynamics of HIV-1 under impaired humoral immunity. The paper addresses important aspects of the subject, but there are several concerns that need to be addressed before considering publication. Below are my comments and suggestions:
    Major Concerns:
    Comparative Results and Novelty: The manuscript lacks comparative results to highlight the new achievements proposed in this work in relation to the existing literature. To ensure fairness, the authors must explicitly state whether the results presented here are significantly superior to those found in the available literature. This comparison is essential to demonstrate the novelty and contribution of the current study.
    Reply: In the discussion section, we have added a comparison between our proposed model and the models presented in the literature.
    Applicability and Validation: The authors do not provide evidence of verifying the applicability of the proposed models in clinical practice using real-world data. I strongly recommend validating the fractional models with real experimental data to ensure their relevance and practicality.
    Reply: We have added the following in the discussion section. 
    "The main limitation of of our present work is that we did not use real data to estimate the values of the model's parameters. The main reasons are as follows: (1) The real data from HIV-1 infected individuals are still few; (2) Comparing our obtained results with a few number of real studies may not be very precise; (3) Collecting real data from HIV-1 infected patients is not an easy task; (4) Working on experiments to obtain real data is beyond the scope of this paper. Thus, the theoretical results obtained in this paper need to be tested against empirical findings when real data become available."
    Typos, Grammar, and Figures: Careful proofreading is required to rectify typos and grammatical errors present in the manuscript. Additionally, the figures need thorough checking to ensure they accurately represent the discussed concepts.
    Notably, the absence of the effects of variables like \kappa in figure 3 and \ro in figure 4 should be addressed.
    Reply: We have corrected all Typos, Grammar, and Figures.
    Minor Concerns:
    Inclusive Conclusion: A comprehensive and concise conclusion is crucial to summarize the paper's findings and significance effectively. I encourage the authors to provide a concise yet impactful conclusion that emphasizes the contribution and implications of their work.
    Reply: The discussion, conclusion and future work have been separated into three sections. 
    DOIs for References: Ensure that all references include DOIs for improved accessibility and citation accuracy.
    Reply: We have added the DOIs for References.
    Major Comments for Improvement:
    The paper would benefit from a more detailed explanation of the mathematical models, their components, and their equations. This would aid readers unfamiliar with the subject matter in comprehending the study. Discuss the rationale behind choosing the specific compartments and variables in the models. Clarify why these components are essential for capturing the dynamics of HIV-1 under impaired humoral immunity.
    Reply: We have defind all variables and paramters given in the models. We have also explained the interactions between the HIV-1 particles, target cells and immune cells.
    Provide a detailed description of the methodology used to conduct numerical simulations. Explain the parameters chosen and their significance in reflecting real-world scenarios.
    Reply: We have added the following "We use ode45 and dde23 solvers in MATLAB to perform the numerical simulations for systems (2.1) and (3.1), respectively. Other methods can also be used for solving these systems (see e.g., [50] and [51]).".
    We have also added the following:
    "The values of most parameters are taken from previous studies. The other parameters ρ_{i}, i=1,2,3 will be chosen just to perform the numerical simulation."

Replies to the comments of Reviewer #4

    First we would like to thank the reviewer for reading our paper and for his constructive and valuable comments.
    I would like to suggest some areas for further consideration and improvement in the manuscript
    (Minor Revision)
    Clarity of Presentation: While your paper is technically sound, there are instances where the clarity of presentation could be improved. Consider providing additional explanations for complex mathematical concepts, particularly for readers with a broader scientific background. This will ensure that the work is accessible to a wider audience.
    Reply: We have defind all variables and paramters given in the models. We have also explained the interactions between the HIV-1 particles, target cells and immune cells.
    Discussion of Biological Implications: While your mathematical models offer insights into the dynamics of HIV-1, it would be beneficial to delve deeper into the biological implications of your findings. Discuss how the identified factors (B-cell impairment, time-delay, latent CI) align with existing biological knowledge and provide potential mechanisms underlying these effects.
    Reply: This has been discussed in the discussion section.
    Comparison with Previous Studies: To contextualize your findings within the existing literature, consider providing a brief comparison with previous mathematical models of HIV-1 dynamics. Highlight the novel aspects of your work and how your study contributes to advancing the field.
    Reply: In the discussion section, we have added a comparison between our proposed model and the models presented in the literature.
    Sensitivity Analysis: Given the complex nature of the model and the interplay of various parameters, conducting a sensitivity analysis could further enhance the robustness of your findings. Investigate the sensitivity of key model parameters and assess their impact on the results.
    Reply: We have added subsection 4.3 for sensitivity analysis.
    Practical Implications: Extend the discussion on the practical implications of your findings for the development of HIV-1 treatment strategies. How might your results inform the design of more effective drug therapies or intervention strategies? This would add depth to the potential real-world applications of your research.
    Reply: The discussion has been extended to clarify these points. 
    Minor language modification is required.
    Reply: It has been done.

Reviewer 2 Report

This is an interesting manuscript dealing with two different models of within-host dynamics of human immunodeciency virus type-1 (HIV-1) with impaired humoral immunity. The results are important in uderstanding HIV disease and its dynamics. It would be better if the authors could give physical/biological interpretation of the basic reproduction number and discuss about the implication in clinical practice. Moreover, sensitivity analysis is also needed to show the significance both the parameters, especially those appeared in the basic reproductio number.

Author Response

(The authors gave the same response as above.)

Reviewer 3 Report

Extensive editing of English language required.

Review Report

Title: "Stability of Impaired Humoral Immunity HIV-1 Models with Active and Latent Cellular Infections"

I have carefully reviewed the manuscript titled "Stability of Impaired Humoral Immunity HIV-1 Models with Active and Latent Cellular Infections." I appreciate the authors' efforts in exploring the within-host dynamics of HIV-1 under impaired humoral immunity. The paper addresses important aspects of the subject, but there are several concerns that need to be addressed before considering publication. Below are my comments and suggestions:

Major Concerns:

  1. Comparative Results and Novelty: The manuscript lacks comparative results to highlight the new achievements proposed in this work in relation to the existing literature. To ensure fairness, the authors must explicitly state whether the results presented here are significantly superior to those found in the available literature. This comparison is essential to demonstrate the novelty and contribution of the current study.

  2. Applicability and Validation: The authors do not provide evidence of verifying the applicability of the proposed models in clinical practice using real-world data. I strongly recommend validating the fractional models with real experimental data to ensure their relevance and practicality.

  3. Typos, Grammar, and Figures: Careful proofreading is required to rectify typos and grammatical errors present in the manuscript. Additionally, the figures need thorough checking to ensure they accurately represent the discussed concepts. Notably, the absence of the effects of variables like \kappa in figure 3 and \ro in figure 4 should be addressed.

Minor Concerns:

  1. Inclusive Conclusion: A comprehensive and concise conclusion is crucial to summarize the paper's findings and significance effectively. I encourage the authors to provide a concise yet impactful conclusion that emphasizes the contribution and implications of their work.

  2. DOIs for References: Ensure that all references include DOIs for improved accessibility and citation accuracy.

Major Comments for Improvement:

  1. The paper would benefit from a more detailed explanation of the mathematical models, their components, and their equations. This would aid readers unfamiliar with the subject matter in comprehending the study.

  2. Discuss the rationale behind choosing the specific compartments and variables in the models. Clarify why these components are essential for capturing the dynamics of HIV-1 under impaired humoral immunity.

  3. Provide a detailed description of the methodology used to conduct numerical simulations. Explain the parameters chosen and their significance in reflecting real-world scenarios.

I appreciate the authors' contribution to the field, and I believe addressing these concerns will significantly enhance the manuscript's quality and potential impact. I look forward to reviewing the revised version of the paper.

Author Response

(The authors gave the same response as above.)

Reviewer 4 Report

I would like to suggest some areas for further consideration and improvement in the manuscript

(Minor Revision)

Clarity of Presentation: While your paper is technically sound, there are instances where the clarity of presentation could be improved. Consider providing additional explanations for complex mathematical concepts, particularly for readers with a broader scientific background. This will ensure that the work is accessible to a wider audience.

Discussion of Biological Implications: While your mathematical models offer insights into the dynamics of HIV-1, it would be beneficial to delve deeper into the biological implications of your findings. Discuss how the identified factors (B-cell impairment, time-delay, latent CI) align with existing biological knowledge and provide potential mechanisms underlying these effects.

Comparison with Previous Studies: To contextualize your findings within the existing literature, consider providing a brief comparison with previous mathematical models of HIV-1 dynamics. Highlight the novel aspects of your work and how your study contributes to advancing the field.

Sensitivity Analysis: Given the complex nature of the model and the interplay of various parameters, conducting a sensitivity analysis could further enhance the robustness of your findings. Investigate the sensitivity of key model parameters and assess their impact on the results.

Practical Implications: Extend the discussion on the practical implications of your findings for the development of HIV-1 treatment strategies. How might your results inform the design of more effective drug therapies or intervention strategies? This would add depth to the potential real-world applications of your research.

Minor language modification is required

Author Response

(The authors gave the same response as above.)

Round 2

Reviewer 1 Report

The authors introduced all corrections (not only mine, but also other reviewers). They addressed all comments and suggestions. In my opinion, the work is suitable for printing.

Author Response

Response to Reviewers
Stability of impaired humoral immunity HIV-1 models with active and latent cellular infections

Noura H. AlShamrani , Reham H. Halawani, Wafa Shammakh and Ahmed M. Elaiw

    First we thank the Editor who handling our paper. We also would like to thank the anonymous reviewers for their valuable comments and suggestions. We have revised the paper carefully in accordance with these comments and suggestions.

Replies to the comments of Reviewer #1

    First we would like to thank the reviewer for reading our paper.
    The authors introduced all corrections (not only mine, but also other reviewers). They addressed all comments and suggestions. In my opinion, the work is suitable for printing.
    Reply: Thanks a lot for the reviewer's recommendation

Replies to the comments of Reviewer #3

    First we would like to thank the reviewer for reading our paper and for his constructive and valuable comments.
    I appreciate your responses; however, I still have some reservations regarding the comparison and limitations addressed. While I understand the challenges associated with obtaining real data for HIV-1 infected individuals, I believe that a more comprehensive analysis could strengthen the validity of the proposed model. The comparison with existing literature models is a step in the right direction, but it would be beneficial to have a direct comparison with any available empirical data, even if it's limited in scope. This would provide a basis for assessing the practical applicability of the proposed model.
    Reply: In fact, to estimate the model's parameters we need real data from HIV infected patients with impaired humoral immunity not from any HIV infected patient. Most of the existing data are not given without mentioning whether it is for patients with humoral immunity or not. This is the difficulty of comparing our model with real data. 
    Additionally, I was hoping for more insights into the potential sources of numerical errors or assumptions that might impact the accuracy of the results. Addressing these points would not only enhance the transparency of your findings but also give readers a better understanding of the potential limitations they should be aware of when interpreting the results.
    I understand the challenges inherent in obtaining real data, but given the importance of empirical validation in scientific research, I'm concerned that relying solely on theoretical results might limit the broader impact of your conclusions. Could you please provide further clarification or context on these aspects?"
    Reply: In our paper we did not developed a new numerical method for solving the ordinary or delay differential equations because this is not our objective in this paper. We just used efficient and popular numerical solvers ode45 and dde23 presented in MATLAB which are based on the Runge-Kutta method with variable step size. ode23 is a three-stage, third-order, Runge-Kutta method. ode45 is a six-stage, fifth-order, Runge-Kutta method. The errors in these methods are very small and does not affect on the stability of the equilibria.

Reviewer 3 Report

I appreciate your responses; however, I still have some reservations regarding the comparison and limitations addressed. While I understand the challenges associated with obtaining real data for HIV-1 infected individuals, I believe that a more comprehensive analysis could strengthen the validity of the proposed model. The comparison with existing literature models is a step in the right direction, but it would be beneficial to have a direct comparison with any available empirical data, even if it's limited in scope. This would provide a basis for assessing the practical applicability of the proposed model.

Additionally, I was hoping for more insights into the potential sources of numerical errors or assumptions that might impact the accuracy of the results. Addressing these points would not only enhance the transparency of your findings but also give readers a better understanding of the potential limitations they should be aware of when interpreting the results.

I understand the challenges inherent in obtaining real data, but given the importance of empirical validation in scientific research, I'm concerned that relying solely on theoretical results might limit the broader impact of your conclusions. Could you please provide further clarification or context on these aspects?"

Moderate editing of English language required

Author Response

(The authors gave the same response as above.)
